# Mean Phase Voltages and Duty Cycles Estimation of a Three-Phase Inverter in a Drive System Using Machine Learning Algorithms

Nikola Anđelić, Ivan Lorencin *, Matko Glučina and Zlatan Car

Faculty of Engineering, University of Rijeka, 51000 Rijeka, Croatia
* Correspondence: ilorencin@riteh.hr

**Abstract:** To achieve an accurate, efficient, and high dynamic control performance of electric motor drives, precise phase voltage information is required. However, measuring the phase voltages of electrical motor drives online is expensive and potentially contains measurement errors, so they are estimated by inverter models. In this paper, the idea is to investigate if various machine learning (ML) algorithms could be used to estimate the mean phase voltages and duty cycles of the black-box inverter model and black-box inverter compensation scheme with high accuracy using a publicly available dataset. Initially, nine ML algorithms were trained and tested using default parameters. Then, the randomized hyper-parameter search was developed and implemented alongside a 5-fold cross-validation procedure on each ML algorithm to find the hyper-parameters that will achieve high estimation accuracy on both the training and testing part of a dataset. Based on obtained estimation accuracies, the eight ML algorithms from all nine were chosen and used to build the stacking ensemble. The best mean estimation accuracy values achieved with stacking ensemble in the black-box inverter model are $\overline{R}^2 = 0.9998, \overline{MAE} = 1.03$, and $\overline{RMSE} = 1.54$, and in the case of the black-box inverter compensation scheme $\overline{R}^2 = 0.9991, \overline{MAE} = 0.0042$, and $\overline{RMSE} = 0.0063$, respectively.

**Keywords:** duty cycle; electrical drive; machine learning; phase voltages; three-phase inverter



## 1. Introduction

A power inverter is a power electronic device that converts direct current (DC) to alternating current (AC). Such power electronic circuits are used to convert one power source waveform into another, in this case from DC to AC. Power electronics have a key role in the conversion of waveforms of power sources and management of renewable energy sources systems [1]. The design of a power inverter dictates input and output voltage, frequency, and overall power handling. A stable DC source that provides enough power for the entire system is required for the proper functioning of the power inverter.

The power inverter can be classified based on the input voltage, so 12 V is the most common voltage used for smaller consumer and commercial inverters, 24–48 V are common for home energy systems, 200 to 400 V for photovoltaic solar panels, 300–450 V for electric vehicle battery packs, and hundreds to thousands of volts for high-voltage power transmission systems.

The output waveform of a power inverter can be a square wave [2], sine wave [3], pulsed sine wave [4], or pulse-width-modulated wave (PWM) [5,6]. Common types of power inverters produce square waves or quasi-square waves [7], where the output frequency of AC is usually 50 or 60 Hz. However, in power inverter designs used for a motor, driving the variable frequency results in variable speed control. The output voltage is usually regulated to be the same as the grid line voltage, even when there are changes in the load that the inverter is driving. Today, power inverters are used in various devices such as converters of DC from batteries or fuel cells to AC [8], uninterruptible power supply (UPS) [9–11], electric motors for producing a variable output voltage range to regulate the

speed of an electrical drive [12–14], refrigeration/air conditioning [15–17] for compressor motor speed to drive refrigerant flow in refrigeration, which results in regulation of system performance, power grid as grid-tied inverters [18–21] to feed into the electric power system and synchronverters [22–25] to simulate a rotating generator for grid stabilization, and convert low-frequency main AC power to higher frequencies in the induction heating process [26–29].

To achieve accurate, efficient, and highly dynamic control performance of electric motor drives, precise phase voltage information is required. This requirement is mandatory for the dynamic control of electric drive performance where torque-controlled operation is considered.

Today, most electric drives do not measure the phase voltages online, so they have to be estimated with inverter models. However, estimation is not accurate due to various non-linear switching effects that occur on a nanosecond scale, and the application of analytical (white box) modeling is hardly feasible. The challenge could be solved with the application of Artificial Intelligence (AI) algorithms that require data acquisition from inverter models.

The development and advancement of power inverters can be achieved with the application of AI algorithms. Several research papers listed below show some advancements in the field of power inverters with the application of AI. Fuzzy logic has been successfully applied in [30,31] for speed control of motor systems with inverters. The normal model of an inverter–induction motor combination and a vast range of faulted models have been developed in [32] using a generic commercial simulation tool to generate voltages and current signals at a broad range of operating points chosen using an ML algorithm. The structured neural network system has been developed to detect and isolate the most common types of faults of inverter–motor combination: single switch open circuit faults, post-short circuits, short circuits, and the unknown faults. Countless conducted system simulations showed that neural network system trained using a machine learning approach achieves high accuracy in detecting whether a faulty condition occurred. The genetic algorithm was applied in [33,34] for harmonic minimization in multilevel inverters. In [35], the authors have utilized the hybrid AI technique of the neuro-genetic algorithm for condition monitoring, fault diagnosis, and the evaluation of an induction motor without any additional information. The term neuro-genetic algorithm is the combination of a backpropagation neural network and genetic algorithm. In [36], the authors have developed a comprehensive AI framework for the fast and reliable classification of distributed generation units' islanding and non-islanding events, with the focus on practical limitations and requirements of a smart power electronics inverter as the desirable observational site.

In this paper, the idea is to investigate the possibility of utilizing a complex ML ensemble system (stacking ensemble) to estimate the mean phase voltages and duty cycles of a three-phase IGBT two-level inverter for electrical drives. The ML ensemble system is a technique that combines basic ML algorithms to produce one optimal estimation model. So, the idea is to develop an optimal estimation model using an ML ensemble system that can estimate mean phase voltages and duty cycles. Generally, two different models were considered:

- Black-box inverter model;
- Black-box inverter compensation scheme.

Here, the term "black-box" refers to the utilization of complex ML algorithms. In the black-box inverter model, the goal is to obtain ML algorithms that could estimate the mean phase voltages of the inverter with high accuracy. In a black-box inverter compensation scheme, the goal is to obtain ML algorithms that could estimate the duty cycles of the inverter with high accuracy. To build this complex ensemble, system selection of basic ML algorithms is required in specific steps. The first step is to investigate which one of the available ML algorithms can achieve good estimation accuracy with default parameters, which can be described as an "out of the box" approach. The second step is to investigate the randomized hyper-parameter search with cross-validation to see if estimation accuracies could be improved. Finally, those ML algorithms that achieved the highest accuracies are

randomly selected in the training process of ensemble methods to see which combination of estimators achieves the highest estimation accuracy. To summarize, the hypotheses of this research are:

- Is it possible to achieve high estimation accuracies of black-box inverter models and a black-box inverter compensation scheme targeted variables (to clarify, the targeted variables in the black-box inverter model are mean phase voltages $\overline{u}_{x,k-1}$, $x \in a, b, c$, while the targeted variables in the case of the black-box inverter compensation scheme are duty cycles at $k-2$ sample $d_{x,k-2}$, $x \in a, b, c$), using different ML algorithms with default parameters;
- Is it possible to improve the estimation accuracies of black-box inverter models and a black-box inverter compensation scheme targeted variables, with a randomized hyper-parameter search with 5-fold cross-validation applied on ML algorithms used in the previous step;
- Is it possible to develop the stacking ensemble (using ML algorithms that achieved the highest estimation accuracies in the previous step) and, on that stacking ensemble, apply the randomized hyper-parameter search with 5-fold cross-validation to achieve high estimation accuracy with improved generalization and robustness of targeted variables in the black-box inverter model and black-box inverter compensation scheme.

The structure of this paper can be divided into the following sections, i.e., Materials and Methods, Results, Discussion, and Conclusions. In the Materials and Methods section, research methodology, dataset description with statistical and correlation analysis, utilized ML algorithms, 5-fold cross-validation procedure with randomized hyper-parameter search, and stacking ensemble methods are described. In the Results section, the results of an initial investigation are presented as well as the results of a randomized hyper-parameter search in combination with a 5-fold cross-validation and stacking ensemble are presented. In the Discussion section, the results obtained in the previous section are discussed. In the Conclusions section, the conclusions are given based on the hypotheses defined in this section and the Results and Discussion sections.

## 2. Materials and Methods

In this section, dataset description, research methodology, various ML algorithms, randomized hyper-parameter search with cross-validation, stacking ensemble, and evaluation methodology are described.

### 2.1. Dataset Description

The dataset description subsection is divided into system description and statistical analysis. The system description is the detailed description of the system, which authors in [37] used to develop the dataset. In the sub-subsection "statistical analysis", the results of statistical analysis and correlation analysis applied to the dataset are presented.

### 2.1.1. System Description

In this paper, a publicly available dataset [37] was utilized, which was obtained using data acquisition and measurement techniques of various signals of a system that consists of a 3-phase power inverter that is a part of the control system for an induction motor. In addition to the inverter and an induction motor, the system consists of DC-link, capacitor stabilizer, digital control system, sensors, and induction motor. The schematic overview of the system is presented in Figure 1.

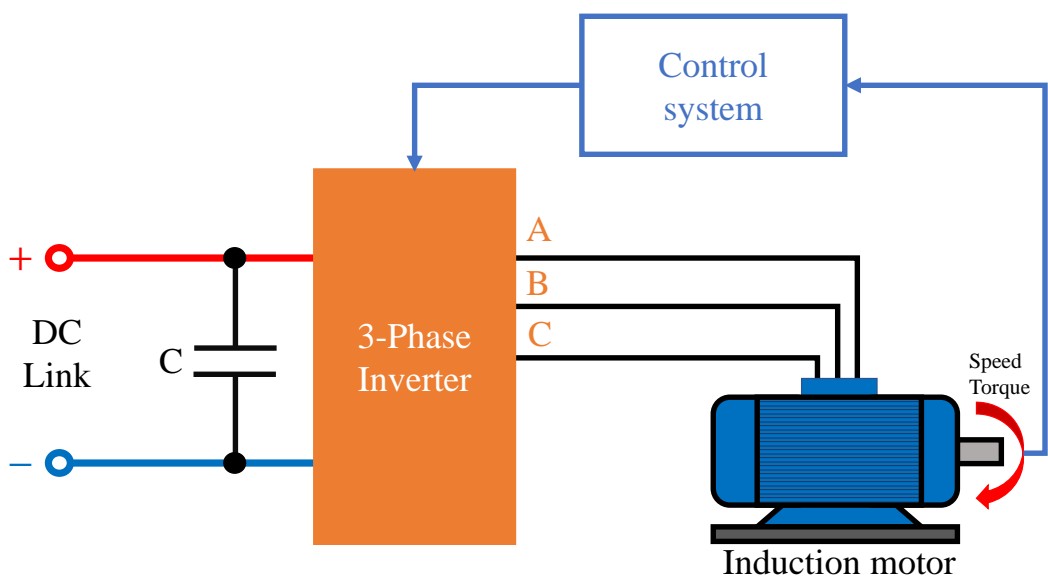

**Figure 1.** Graphical representation of the drive system.

As seen from Figure 1, the induction motor (LUST ASH-22-20K13-000) is fed by a three-phase two-lever IGBT inverter (SEMIKRON Semiteach IGBT), the schematic view of which is shown in Figure 2.

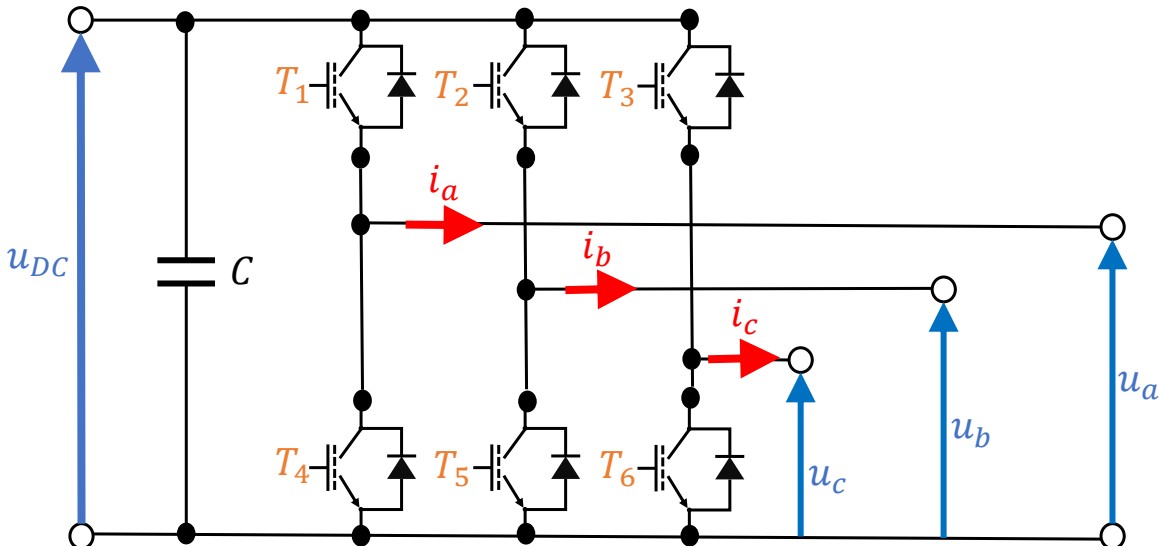

**Figure 2.** Schematics of the inverter in the drive system.

A digital control system performs the control of the motor by the use of field-oriented control (FOC) with a switching frequency of 10 kHz. This type of system can be used for scalar and vector control of an induction motor. The induction motor control is achieved by changing the voltage and frequency of the 3-phase inverter. The change in the voltage and frequency is achieved with sinusoidal PWM. The PWM pulses are supplied to the gates of the insulated-gate bipolar transistor (IGBT) of the inverter. The measurement structure used by the authors in [37] is shown in Figure 3.

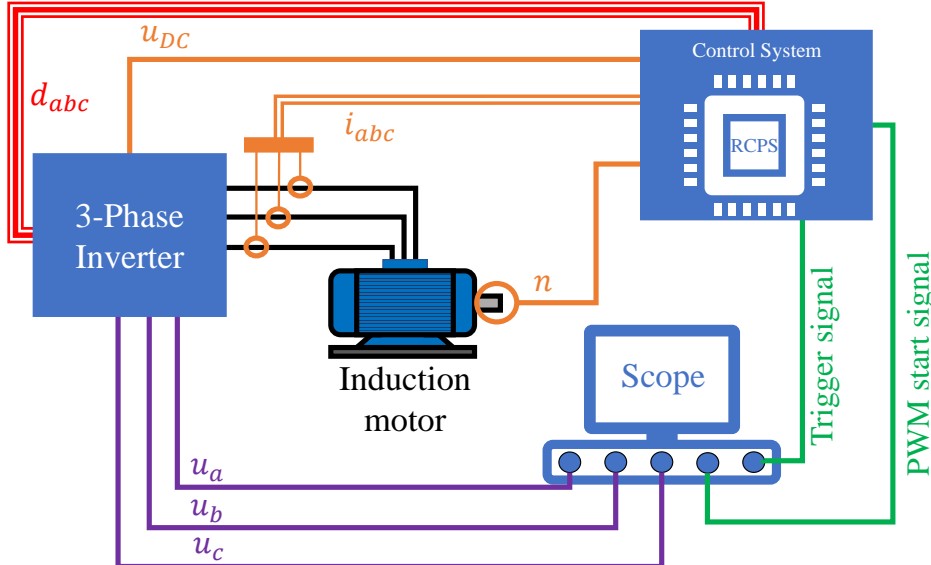

**Figure 3.** Measurement structure.

As seen, Figure 3 consists of a scope (Teledyne LeCroy 12-bit HDO4104), differential probes (1xPMK Bublebee 1 kV CAT III, 2xTeledyne LeCroy ADP305), phase-current sensors (3xSENSITEC CMS2015 SP10), and digital control system (dSPACE MicroLabBox). The pulsating phase voltages $u_a, u_b$, and $u_c$ were measured simultaneously using scope and three differential probes. To identify mean phase voltages at each PWM interval, the trigger signal and signal that indicates a new PWM interval were used to synchronize the rapid control prototyping system (RCPS) and the scope. The voltages were recorded by the scope, and all other signals were recorded by the RCPS at any operating point.

To capture measurement data in complete current and voltage range, the authors in [37] have connected an induction motor to another motor that was speed-controlled by an industrial inverter and control unit. By doing so, the induction motor is operated in current-control mode, which enables the measurement of various duty cycles and current amplitude combinations. The system in this configuration enables data acquisition of samples in steady-state and dynamic operation at various motor speeds.

2.1.2. Dataset Statistical Analysis

The dataset [37] consists of phase voltages, phase currents, duty cycles, DC-link voltage, and speed for 235 thousand sampling steps. Generally, the signals can not be interpreted as continuous measurement records since the dataset is a sequence of several small recorded sequences. So, required past values of the signals for training the inverter models and compensation scheme must be included as additional signals in the dataset for each sampling step.

As already stated, two different models are analyzed: the black-box inverter model and the black-box inverter compensation scheme. The targeted output in the black-box inverter model are mean phase voltages $\overline{u}_{x,k-1}$ of phases $x \in a, b, c$, while the targeted output in case of the black-box inverter compensation is the duty cycles $d_{x,k-2}$ of phase $x \in a, b, c$. The duty cycles $d_{x,k-2}$ are calculated by a rapid control prototyping system during the PWM period $[k-2, k-1] \cdot T_s$ and are based on the measured phase currents $ix, k-2, x \in [a, b, c]$, and the DC-link voltage during that PWM period, and is set by the inverter during the following PWM period $[k-1, k] \cdot T_s$. In Tables 1 and 2 the results of the statistical analysis (mean values, standard deviation, and minimum and maximum values) of input and output variables are presented.

**Table 1.** Statistical description of the black-box inverter model. (The *k* refers to sampling step).

| Model | Variable Name | Symbol | Mean | STD | Min | Max |
|---|---|---|---|---|---|---|
| Input Variables | Phase Currents | $i_{a,k}$ | 0.0005 | 2.19 | −7.3 | 7.47 |
| | | $i_{b,k}$ | −0.0076 | 2.1553 | −6.3202 | 6.6681 |
| | | $i_{c,k}$ | −0.0089 | 2.2162 | −7.1129 | 7.4371 |
| | Phase Currents at $k-1$ | $i_{a,k-1}$ | 0.0005 | 2.1992 | −7.3001 | 7.4702 |
| | | $i_{b,k-1}$ | −0.0077 | 2.1553 | −6.3202 | 6.6681 |
| | | $i_{c,k-1}$ | −0.0089 | 2.2161 | −7.1129 | 7.4371 |
| | Duty cycles at $k-2$ | $d_{a,k-2}$ | 0.5002 | 0.2119 | 0 | 1 |
| | | $d_{b,k-2}$ | 0.5002 | 0.2117 | 0 | 1 |
| | | $d_{c,k-2}$ | 0.5001 | 0.2117 | 0 | 1 |
| | Duty cycles at $k-3$ | $d_{a,k-3}$ | 0.5002 | 0.2119 | 0 | 1 |
| | | $d_{b,k-3}$ | 0.5002 | 0.2117 | 0 | 1 |
| | | $d_{c,k-3}$ | 0.5001 | 0.2117 | 0 | 1 |
| | DC-link voltage at $k$ | $u_{dc,k}$ | 567.13 | 4.9936 | 548.01 | 575.55 |
| | DC-link voltage at $k-1$ | $u_{dc,k-1}$ | 567.13 | 4.9934 | 548.01 | 575.55 |
| Output Variables | Mean phase voltages at $k-1$ | $\overline{u}_{a,k-1}$ | 283.41 | 114.64 | −2.2884 | 573.34 |
| | | $\overline{u}_{b,k-1}$ | 283.46 | 114.29 | −2.0879 | 573.2 |
| | | $\overline{u}_{c,k-1}$ | 283.74 | 114.6 | −2.3124 | 573.17 |

**Table 2.** Statistical description of the black-box inverter compensation scheme.(The *k* refers to sampling step).

| Model | Variable Name | Symbol | Mean | STD | Min | Max |
|---|---|---|---|---|---|---|
| Input Variables | Mean phase voltages at $k-1$ | $\overline{u}_{a,k-1}$ | 283.41 | 114.65 | −2.2884 | 573.33 |
| | | $\overline{u}_{b,k-1}$ | 283.46 | 114.29 | −2.0879 | 573.2 |
| | | $\overline{u}_{c,k-1}$ | 283.74 | 114.6 | −2.31 | 573.17 |
| | Duty cycles at k-3 | $d_{a,k-3}$ | 0.5002 | 0.212 | 0 | 1 |
| | | $d_{b,k-3}$ | 0.5002 | 0.2117 | 0 | 1 |
| | | $d_{c,k-3}$ | 0.5001 | 0.2117 | 0 | 1 |
| | Phase currents at $k-3$ | $i_{a,k-3}$ | 0.0005 | 2.1989 | −7.3 | 7.47 |
| | | $i_{b,k-3}$ | −0.0078 | 2.1551 | −6.32 | 6.6681 |
| | | $i_{c,k-3}$ | −0.0088 | 2.216 | −7.1129 | 7.4371 |
| | Phase currents at $i-2$ | $i_{a,k-2}$ | 0.0005 | 2.199 | −7.3 | 7.47 |
| | | $i_{b,k-2}$ | −0.0077 | 2.1552 | −6.32 | 6.6681 |
| | | $i_{c,k-2}$ | −0.0089 | 2.2161 | −7.1129 | 7.4371 |
| | DC-link voltage at $k-3$ | $u_{dc,k-3}$ | 567.13 | 4.9931 | 548.01 | 575.5533 |
| | DC-link voltage at $k-2$ | $u_{dc,k-2}$ | 567.13 | 4.9933 | 548.01 | 575.55 |
| Output Variables | Duty cycles at $k-2$ | $d_{a,k-2}$ | 0.5 | 0.2119 | 0 | 1 |
| | | $d_{b,k-2}$ | 0.5002 | 0.2117 | 0 | 1 |
| | | $d_{c,k-2}$ | 0.5001 | 0.2117 | 0 | 1 |

As seen in Tables 1 and 2, a total of 6 different output variables are estimated using different ML regression algorithms. Three variables in the black-box inverter model and three

in the black-box inverter compensation scheme. For each target value, the chosen ML algorithms are trained and tested. The duty cycle depends on the time $T_s$, i.e., on the switching on or off of the IGBT switch in a certain time interval and the target waveform; the voltage values $\overline{u}_{a,k-1}$, $\overline{u}_{b,k-1}$, and $\overline{u}_{c,k-1}$ must be approximately similar in the positive and negative half-cycle. As seen from both tables, the range between minimum and maximum values differs from variable to variable. Initially, the application of scaling and normalization techniques on the dataset was considered, however, high accuracy was achieved with ML algorithms on the original dataset, so the application of data preprocessing methods was omitted from further investigations.

In addition to the initial statistical analysis, it is good practice to investigate the correlation between input and output variables, i.e., perform correlation analysis. Here, Pearson's correlation analysis was applied. The correlation between any input variable and the output variable is in the range of $-1.0$ to $1.0$. The $-1.0$ correlation value between input and output variables means that if the input value decreases, the output value increases, and vice versa. On the other hand, the correlation value of $1.0$ between the input and output variables means that if the input variable increases, the output variable also increases. It should be noted that the worst correlation values are in the range of $-0.5$ to $0.5$, especially $0$, which means that variation of input variable value will have absolutely no effect on the output variable. In Figures 4 and 5, the results of Pearson's correlation analyses are shown for both models in the form of a correlation heat-map.

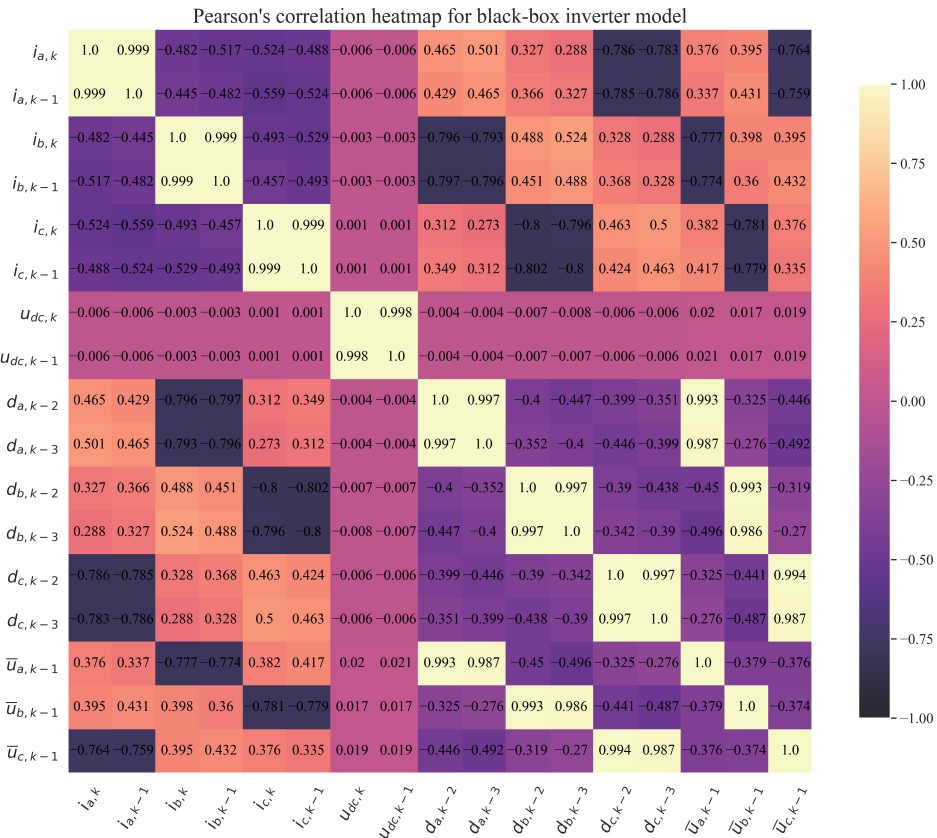

**Figure 4.** Pearson's correlation heat-map for black-box inverter model.

As seen in Figure 4, the three targeted output variables, i.e., mean phase voltages at $k-1$ sample ($\overline{u}_{a,k-1}$, $\overline{u}_{b,k-1}$, and $\overline{u}_{c,k-1}$), has a good correlation with the majority of input variables. However, the correlation with the $u_{dc,k}$ and $u_{dc,k-1}$ variables is near 0, so the initial presumption is that these variables will not have any influence on their estimation. The mean phase voltages at $k-1$ sample have the highest correlation (0.99), with duty cycles at the $k-2$ sample ($d_{a,k-2}$, $d_{b,k-2}$, and $d_{b,k-2}$) and duty cycles with $k-3$ sample

$(d_{a,k-3}, d_{b,k-3},$ and $d_{c,k-3})$, while the lowest negative correlation is with phase currents $(i_{a,k}, i_{b,k},$ and $i_{c,k})$ and with phase currents at $k-1$ sample $(i_{a,k-1}, i_{b,k-1},$ and $i_{c,k-1})$.

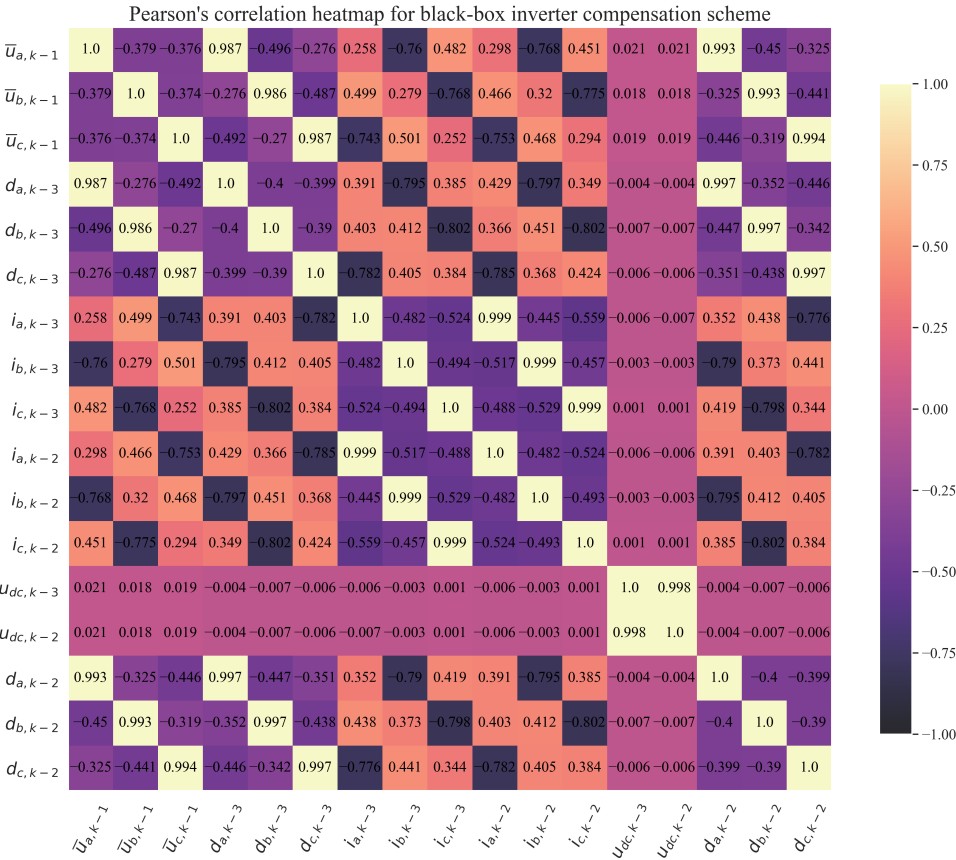

**Figure 5.** Pearson's correlation heat-map for black-box inverter compensation scheme.

The correlation heat-map shown in Figure 4 shows that targeted output variables, i.e., duty cycles at the $k-2$ sample $(d_{a,k-2}, d_{b,k-2},$ and $d_{b,k-2})$, have a good correlation with the majority of input variables, however, there is not any correlation with DC-link voltage at $k-2$ sample $(u_{dc,k-2})$ and $k-3$ $(u_{dc,k-3})$, respectively. The three duty cycles at $k-2$ sample variables have the highest correlation $(\approx 1.0)$ to duty cycles at $k-3$ sample $(d_{a,k-3}, d_{b,k-3},$ and $d_{c,k-3})$ and mean phase voltages at $k-1$ sample $(\overline{u}_{a,k-1}, \overline{u}_{b,k-1}, \overline{u}_{c,k-1})$ while the lowest negative correlation $(\approx -0.8)$ is to phase currents at $k-3$ sample, and k-2 sample, respectively. However, in each model during the training of different ML algorithms, all input variables will be included.

It should be noted that these black-box inverter models were created for two reasons. The authors in [38] have suggested these models (the combination of input and output variables). This suggestion of data configuration for both models was tested and verified by conducting Pearson's correlation investigation, which showed an excellent correlation between input and output variables and can be seen for each model in Figures 4 and 5, respectively. However, the investigation showed that each black-box inverter model has two variables with the lowest correlation to the target variables. As already stated, in the black-box inverter model, the DC-link voltage at $k$ $(u_{dc,k})$ and DC-link voltage at $k-1$ $(u_{dc,k-1})$ have the lowest correlation $(\approx 0)$ with target variables $(\overline{u}_{a,k-1}, \overline{u}_{b,k-1},$ and $\overline{u}_{c,k-1})$. In the case of the black-box inverter compensation scheme, the DC-link voltage at $k-2$ and $k-3$ do not have any correlation with target variables $(d_{a,k-2}, d_{b,k-2},$ and $d_{c,k-2})$. The DC-link voltages at $k, k-1, k-2,$ and $k-3$ were suggested by the authors in [38], so the results of the correlation analysis for these variables were left in Figures 4 and 5. The dataset also contains the speed ([ref/min]) at $k$. However, correlation analysis showed this variable does not have any correlation $(\approx 0)$ with the target variables in both models.

## 2.2. Research Methodology

In this paper, the idea is to investigate if an ML algorithm or ensemble consisting of different ML algorithms could estimate the mean phase voltages of the black-box inverter model and duty cycles of the black-box inverter compensation scheme with high accuracy. The procedure of this research can be summarized in the following steps:

- Perform the initial investigation using the original dataset with various ML algorithms with default parameters to select only those that achieve reasonable high accuracy in mean phase voltages and duty cycle estimation.
- On selected ML regression algorithms, perform a randomized hyper-parameter search with 5-fold cross-validation to find which combination of hyper-parameters for each ML algorithm achieves the highest estimation accuracies.
- Select ML algorithms that achieved the highest estimation accuracies in previous steps to build a stacking ensemble and to investigate if even higher estimation accuracies could be achieved.

The graphical representation of the research methodology is shown in Figure 6.

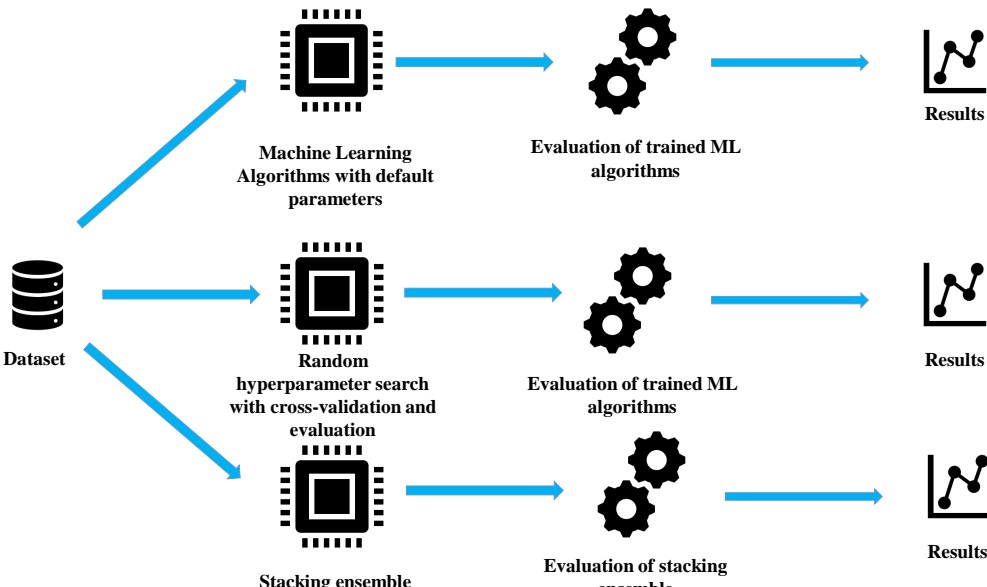

**Figure 6.** Graphical representation of research methodology procedure.

## 2.3. Various Machine Learning Algorithms

In this paper, the Python programming language was utilized with the scikit-learn library version 1.0.2 (December 2021). The library contains all the ML algorithms that were used in this investigation. Here, only a basic description of each ML algorithm used is given, and they are Automatic Relevance Determination (ARD), Bayesian Ridge, Elastic Net, Huber, K-Neighbours, Lasso, Linear, Multi-Layer Perceptron (MLP), and Ridge. Each description of the ML algorithm is accompanied by a range of hyper-parameters that were used in their randomized hyper-parameter search during the training of the ML algorithms. It should be noted that not all hyper-parameters are shown in the following tables. However, only the hyper-parameters that were used in the randomized hyper-parameter search are shown and described.

### 2.3.1. ARD Regression Algorithm

The Automatic Relevance Determination regression algorithm [39,40] (ARD) is a linear model, similar to Bayesian Ridge Regression. In this algorithm, the weights of the model are assumed to be in Gaussian distributions. The precisions of the weights

distributions (lambda) and the precision of the noise distribution (alpha) are estimated, which is performed by iterative procedures (Evidence Maximization).

The list of hyper-parameters with a predefined range is given in Table 3.

**Table 3.** An overview of the used hyper-parameter range for ARD regression method.

| Parameter Name | Lower Bound | Upper Bound |
|---|---|---|
| *n_iter* | 100 | 1000 |
| *tol* | $1 \times 10^{-30}$ | $1 \times 10^{-25}$ |
| *alpha_1* | $1 \times 10^{-20}$ | $1 \times 10^{-1}$ |
| *alpha_2* | $1 \times 10^{-20}$ | $1 \times 10^{-1}$ |
| *lambda_1* | $1 \times 10^{-20}$ | $1 \times 10^{-1}$ |
| *lambda_2* | $1 \times 10^{-20}$ | $1 \times 10^{-1}$ |
| *compute_score* | True, False | |
| *threshold_lambda* | 1000 | 100,000 |

In Table 3, *n_iter* is the maximum number of iterations that will be used when training the algorithm. *tol* will stop the algorithm if $w$ parameter values have converged, i.e., dropped below *tol* value. *alpha_1* is a shape parameter for the Gamma distribution prior over the alpha parameter. *alpha_2* is the inverse scale parameter, i.e., the rate parameter for the Gamma distribution is prior over the alpha parameter. *lambda_1* is the shape parameter for the Gamma distribution which is over the lambda parameter. *lambda_2* is the inverse scale parameter, i.e., the rate parameter for the Gamma distribution which is prior over the lambda parameter. If *compute_score* is set to True, then the objective function will be computed at each step of the model. *treshold_lambda* is a threshold for removing weights with high precision from the computation.

### 2.3.2. Bayesian Ridge Regression Algorithm

The Bayesian Ridge Regression Algorithm [40,41] includes regularization parameters in the estimation procedure, and these are tuned to the data at hand. The fully probabilistic model is achieved by assuming the $y$ as Gaussian distribution around $Xw$:

$$p(y|X, w, \alpha) = N(y|Xw, \alpha) \tag{1}$$

where $\alpha$ is a random variable that is estimated from the data. In Bayesian Ridge Regression prior to the coefficient, $w$ is given with spherical Gaussian:

$$p(w|\lambda) = N(w|0, \lambda^{-1} I_p) \tag{2}$$

The parameters $w, \alpha$, and $\lambda$ are estimated jointly during the training of an estimator. The list of hyper-parameters with a predefined range is given in Table 4.

In Table 4, *num_iter* is the maximum number of iterations that the algorithm will execute. The *tol* hyper-parameter will stop the algorithm execution if $w$ has converged. *alpha_1* is a shape parameter for the Gamma distribution before the alpha hyper-parameter. *alpha_2* is the inverse scale parameter or rate parameter for the Gamma distribution prior over the alpha parameter. *lambda_1* is the shape parameter for the Gamma distribution prior before the lambda parameter. *lambda_2* is the inverse scale parameter or rate parameter for the Gamma distribution prior over the lambda parameter. *alpha_init* is the initial value for alpha, i.e., when *None*, the *alpha_init* is equal to $\frac{1}{Var(y)}$. *lambda_init* is the initial value of lambda, i.e., if it was set to *None* the *lambda_init* is equal to 1. *compute_score* is a Boolean type of hyper-parameter. If *True*, then it will compute the log marginal likelihood at each iteration of the optimization. *fit_itercept* is another Boolean-type hyper-parameter.

If *True*, it will calculate the intercept for this model. If *False*, the interception will not be used in calculations, which means that data is expected to be centered.

**Table 4.** An overview of the used hyper-parameter range for Bayesian Ridge Regression Algorithm.

| Parameter Name | Lower Bound | Upper Bound |
|---|---|---|
| *num_iter* | 500 | 1000 |
| *tol* | $1 \times 10^{-4}$ | $1 \times 10^{-3}$ |
| *alpha_1* | $1 \times 10^{-5}$ | $1 \times 10^{-1}$ |
| *alpha_2* | $1 \times 10^{-5}$ | $1 \times 10^{-1}$ |
| *lambda_1* | $1 \times 10^{-5}$ | $1 \times 10^{-1}$ |
| *lambda_2* | $1 \times 10^{-5}$ | $1 \times 10^{-1}$ |
| *alpha*_init | None | $(0, 10)$ |
| *lambda*_init | None | $(0, 10)$ |
| *compute_score* | True, False | |
| *fit_intercept* | True, False | |

### 2.3.3. ElasticNet Regression Algorithm

The ElasticNet regression algorithm [40,42] is a regularized regression method that linearly combines $L1$ and $L2$ penalties of lasso and ridge regression methods. This method is useful in the case where there are multiple correlated features. The difference between the ElasticNet and Lasso is that Lasso is likely to pick one of these features, while the ElasticNet is likely to pick both at once. The ElasticNet is trained with both the $L1$ and $L2$-norm for regularization of the coefficient. This combination allows the learning of a sparse model where few of the weights are non-zero, while still maintaining the regularization properties of the Ridge regularization method. The list of hyper-parameters with predefined values range that was used in the randomized hyper-parameter search is shown in Table 5.

**Table 5.** An overview of the used hyper-parameter range for ElasticNet Regression Algorithm.

| Parameter Name | Lower Bound | Upper Bound |
|---|---|---|
| *alpha* | $-10$ | 10 |
| *l1_ratio* | 0 | 1 |
| *max_iter* | 10,000 | 100,000 |
| *tol* | $1 \times 10^{-30}$ | $1 \times 10^{-5}$ |
| *random_state* | 0 | 50 |
| *selection* | cyclic, random | |

In Table 5, *alpha* is the constant that multiplies the ratio, $\frac{L1}{L2}$ is the tuning parameter that decides how much we want to penalize the model. *l1_ratio*, called the ElasticNet parameter, can be in the range 0–1. If $l1_ratio$ is 1, the penalty would be L1, and if 0, the penalty would be l2. If the value of the *l1_ratio* is between 0 and 1, then the penalty would be a combination of $L1$ and $L2$. *fit_intercept* is a Boolean type of hyper-parameter, and if set to *True*, then the constant is specified, which is added to the decision function, otherwise no intercept will be used in calculations. *max_iter* represents the maximum number of iterations taken for conjugate gradient solvers. *tol* represents the tolerance for the optimization. This value and the updates are compared, and if found updates are smaller than *tol*, the optimization checks the dual gap for optimality and continues until it is smaller than *tol*. *random_state* is the seed of the pseudo-random number generated, which

is used while shuffling data. The *selection* hyper-parameter can be cyclic (features looping over sequentially) or random (random coefficients will be updated in every iteration).

### 2.3.4. Huber Regression Algorithm

According to [40,43], the Huber regression algorithm is a linear regression model that is robust to outliers. This is achieved by applying the linear loss to samples that are classified as outliers. The list of hyper-parameters with predefined values range that was used in the randomized hyper-parameter search is shown in Table 6.

**Table 6.** An overview of the used hyper-parameter range for Huber Regression Algorithm.

| Parameter Name | Lower Bound | Upper Bound |
|:---:|:---:|:---:|
| *epsilon* | 1.1 | 10 |
| *max_iter* | 10,000 | 100,000 |
| *alpha* | $1 \times 10^{-10}$ | $1 \times 10^{-3}$ |
| *fit_itercept* | True, False | |
| *tol* | $1 \times 10^{-30}$ | $1 \times 10^{-10}$ |

In Table 6, *epsilon* controls the number of samples that should be classified as outliers. The smaller the epsilon, the more robust it is to outliers. *max_iter* is the maximum number of iterations that the algorithm should run for. If this value is reached or the algorithm converges before the maximum number of iterations, the execution of the algorithm will be terminated. *alpha* is the regularization parameter. *fit_intercept* calculates the intercept for this algorithm, i.e., data are not centered. If False, data are already centered around the origin. *tol* is the tolerance value, and it will stop the execution of an algorithm if $\max(|pg_i|) <= tol \quad i = 1, \ldots, n$, where $pg_i$ is the *i*-th component of the projected gradient.

### 2.3.5. K-Neighbors Regression Algorithm

K-neighbors regression is a non-parametric supervised learning algorithm [40,44]. The output of k-NN is the property value for the object. The value is the average of the values of *k*-nearest neighbors. The list of parameters with predefined range is given in the Table 7.

**Table 7.** An overview of the used hyper-parameter range for K-Neighbors Regression Method.

| Parameter Name | Lower Bound | Upper Bound |
|:---:|:---:|:---:|
| *n_neighbors* | 2 | 10 |
| *Weights* | uniform, distance | |
| *algorithm* | auto, ball_tree, kd_tree, brute | |

In Table 7, *n_neighbors* specifies the number of neighbors to use by default for *kneighbors* queries. *kneighbors* is the function that finds the K-neighbors of a point and returns indices of and distances to the neighbors of each point. *weights* can be set to uniform (all points in each neighborhood are weighted equally) or distance (the weight point are calculated by the inverse of the distance, i.e., closer neighbors of a query point will have greater influence than those far away). The *algorithm* hyper-parameter can be set to auto (automatically choose any other type of algorithm), ball tree, k-d tree, and brute.

### 2.3.6. Lasso Regression Algorithm

The Least Absolute Shrinkage and Selection Operator (Lasso) is the regression algorithm that estimates the sparse coefficient [40,45]. This algorithm is an example of a regularized regression, which is one of the approaches to tackle the problem of over-fitting

by the addition of additional information to shrink the parameter values of the model to induce a penalty against complexity. The list of hyper-parameters with predefined values ranges that was used in the randomized hyper-parameter search is shown in Table 8.

**Table 8.** An overview of the used hyper-parameter range for Lasso Regression Algorithm.

| Parameter Name | Lower Bound | Upper Bound |
|:---:|:---:|:---:|
| *alpha* | 0.1 | 10 |
| *fit_intercept* | True, False | |
| *max_iter* | 1000 | 10,000 |
| *tol* | $1 \times 10^{-30}$ | $1 \times 10^{-10}$ |
| *random_state* | 0 | 50 |
| *selection* | cyclic, random | |

In Table 8, *alpha* is the tuning parameter that decides how much the model will be penalized. *fit_itercept* specifies a constant which should be added to the decision function. If *False*, no intercept will be used in the calculation. *tol* is the tolerance for the optimization. The parameter value and updates are compared, and if they are smaller than the predefined value, the optimization checks the dual gap optimally and continues until the dual gap value is smaller than *tol*. *max_iter* is the maximum number of iterations taken for conjugate gradient solvers. *random_state* represents the seed of the pseudo-random number generated, which is used while shuffling the data. *selection* can be set to *cyclic* (features looping over sequentially) or *random* (random coefficient will be updated in every iteration).

### 2.3.7. Linear Regression Algorithm

Linear regression is one of the best statistical methods used to study the relationship between a dependent variable (Y) with a given set of independent variables (X) [40,46]. The relationship is established with the help of fitting the best line. The list of hyper-parameter values used in the randomized hyper-parameter search is shown in Table 9.

**Table 9.** An overview of the used hyper-parameter range for Linear Regression Algorithm.

| Hyper-Parameter | Bounds |
|:---:|:---:|
| *fit_intercept* | True, False |

In Table 9, *fit_intercept* is a Boolean type of hyper-parameter that is used to calculate the intercept for the model. If set to *False*, no intercept will be used in the calculation.

### 2.3.8. Multi-Layer Perceptron

The multi-layer perceptron (MLP) [40,47] is an ML algorithm that during the training process tries to connect a set of inputs (*X*) to the output (*y*). The general structure of MLP consists of three layers, i.e., input, hidden, and the output layer. The input layer consists of a set of neurons $(x_1, x_2, \ldots, x_m)$, which are inputs. The neurons in the hidden layer transform the values from the input layer with weighted linear summation $w_1 x_1 + w_2 x_2 + \ldots + w_m x_m$, followed by a non-linear activation function. The output layer receives the data from the hidden layer and transforms it to the output value. The MLP Regressor trains using backpropagation with no activation function in the output layer. The list of hyper-parameters with a predefined range is given in Table 10.

The number of hidden layers and number of neurons per hidden layer are both parameters required to define *hidden_layer_sizes*, which can be written as $(hl_1, hl_2, \ldots, hl_n)$, where *n* represents the number of hidden layers and the *hl* value represents the number of neurons in the hidden layer. *activation* represents the type of activation function that

will be used in each neuron in all hidden layers (*identity*, *logistic*, *tanh*, and *relu*). *solver* is the type of solver used for weight optimization (*lbfgs*, *sgd*, and *adam*). *lbfgs* is an optimizer in the family of quasi-Newton methods. *sgd* is the stochastic gradient descent solver. *adam* is the stochastic gradient-based optimizer. *alpha* is the strength of the L2 regularization term. *batch_size* represents the size of mini-batches used for the stochastic optimizer. *learning_rate* represents the learning rate schedule for weight updates (*constant*, *invscaling*, and *adaptive*). The *max_iter* value represents the maximum number of iterations. The *tol* h value represents the tolerance for the optimization. If the loss or score is not improving by at least *tol* for the specific number of iterations, convergence is reached and the training stops. *n_iter_no_change* is the maximum number of epochs in which the *tol* value is not improved.

**Table 10.** An overview of the used hyper-parameters range for MLP Regression Algorithm.

| Parameter Name | Lower Bound | Upper Bound |
|---|---|---|
| Number of Hidden Layers | 2 | 5 |
| No. Neurons per Hidden Layer | 10 | 200 |
| *activation* | identity, logistic, tanh, relu | |
| *solver* | lbfgs, sgd, adam | |
| *alpha* | $1 \times 10^{-6}$ | $1 \times 10^{-2}$ |
| *batch_size* | 200 | 300 |
| *learning_rate* | constant, invscaling, adaptive | |
| *max_iter* | 200 | 2000 |
| *tol* | $1 \times 10^{-10}$ | $1 \times 10^{-3}$ |
| *n_iter_no_change* | 10 | 10,000 |

## 2.3.9. Ridge Regression

The ridge regression (Tikhonov regularization) can be described as a linear least squares estimator with L2 regularization. [40,48]. This ML algorithm solves a regression model where the loss function is the linear least squares function and regularization is given by L2-norm. The list of hyper-parameters with pre-defined values range used in the randomized hyper-parameter search is shown in Table 11.

**Table 11.** An overview of the used hyper-parameter range for Ridge Regression Algorithm.

| Hyper-Parameter | Lower Bound | Upper Bound |
|---|---|---|
| *alpha* | 1.0 | 1000 |
| *fit_intercept* | True | False |
| *max_iter* | 100 | 100,000 |
| *tol* | $1 \times 10^{-9}$ | $1 \times 10^{-3}$ |
| *solver* | *auto, svd, cholesky, lsqr, sparse_cg, sag, saga* | |

In Table 11, *alpha* is the tuning parameter that decides how much the model is to be penalized. *fit_intercept* is the constant which is added to the decision function. If set to *False*, no intercept will be used in the calculation. *max_iter* represents the maximum number of iterations taken for conjugate gradient solvers. The *tol* value represents the precision of the solution. *solver* represents the solver that will be used in computational routines, and *auto* , *svd* , *cholesky* , *lsqr* , *sparse_cg*, *sag*, and *saga* are available.

### 2.4. Randomized Hyper-Parameter Search with Cross-Validation

Cross-validation is one of the key methods used for the determination of both regression and generalization performances. In the case of this research, the *k*-fold cross-validation procedure is used. Such a procedure is performed by dividing the training data sets into *k* parts, where one part of the data set is used for testing, while other parts of the data set are used for training [49]. In the case of this research, a 5-fold cross-validation is used. The dataset is initially divided into train and test datasets in the same ratio as in the previous investigations (train-70%, test-30%). Additionally, the training dataset was used to perform the 5-fold cross-validation.

A schematic representation of the used cross-validation procedure is given in Figure 7.

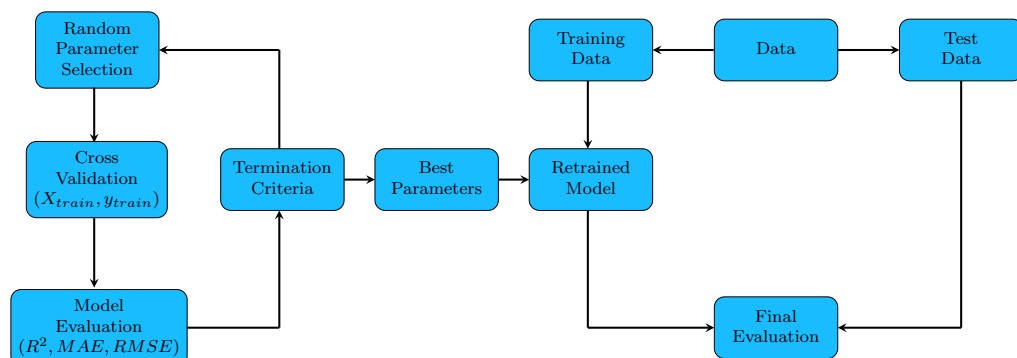

**Figure 7.** The scheme of cross-validation process with randomized hyper-parameter search using various evaluation metrics ($R^2$, $MAE$, and $RMSE$) with final evaluation on the test dataset.

As seen from Figure 7, the dataset is initially divided on the train and test set in the ratio 70:30, where 70% of the dataset is used for cross-validation with randomized hyper-parameter search. In the first loop, the hyper-parameters of the ML algorithm are randomly chosen from a predefined range and used in a 5-fold cross-validation on the training dataset. The performance of the model is evaluated, and average $R^2$, $MAE$, and $RMSE$ values are calculated. In the termination criteria block after each cross-validation, the evaluation metric values are compared with predefined termination criteria values ($R^2 > 0.99$, $MAE < 10.0$, and $RMSE < 12.0$). If obtained averaged values are above ($R^2$) or below ($MAE$ and $RMSE$) predefined values, the cross-validation process with randomized hyper-parameter search is terminated, and if not, new parameters of the ML algorithm are randomly selected, and the process is repeated. If, however, the termination criteria are met, the parameters of the ML algorithm are used to re-train the model on the training dataset, and a final evaluation is performed on the test dataset to obtain $R^2$, $MAE$, and $RMSE$ values.

### 2.5. Stacking Ensemble Method

The idea behind the ensemble method is to combine multiple base estimators to improve generalization and robustness when compared with the performance of a single estimator. In this research, the stacking ensemble method is used. This ensemble learning method seeks a diverse group of members by varying the model types fit on the training data and using a model to combine predictions [50]. In this method, there are usually two levels of models: level-0 and level-1 models. The level-0 are ensemble members, while level-1 is usually one model used to combine the predictions of ensemble members. The structure can be expanded to multiple levels, so, for example, level-1 can have 3 to 5 models, and in level-2 is one model that combines the predictions made by level-1 models. The general schematic overview of the stacking ensemble learning model with two levels (level-0, and level-1) is shown in Figure 8.

The key elements of this model are an unchanged training dataset, different ML algorithms for each ensemble member, and an ML algorithm to learn how to combine predictions. In this paper, the stacking ensemble regression algorithm is used.

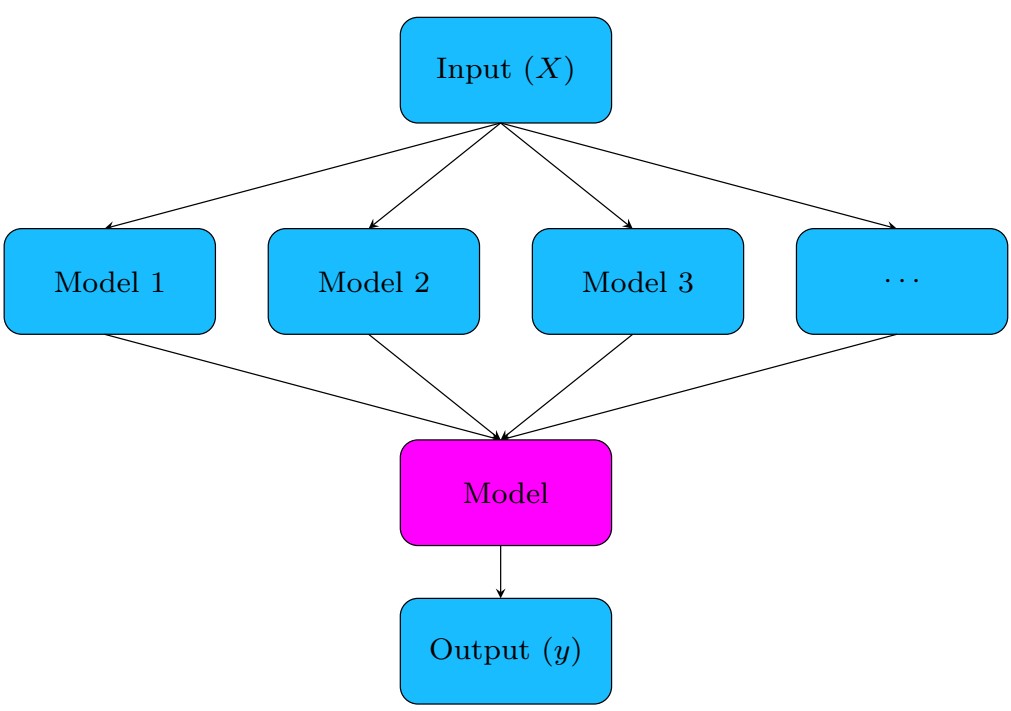

**Figure 8.** Schematic view of stacking ensemble.

*2.6. Evaluation Methodology*

To evaluate the estimation accuracies of all ML algorithms in this paper, three types of evaluation metrics were utilized, i.e.,: coefficient of determination ($R^2$) [51], mean absolute error ($MAE$) [52], and root mean squared error ($RMSE$) [53].

The majority of papers with AI and ML investigation, regardless of training/testing procedure, always show only scores obtained with evaluation methods achieved on test results. However, the evaluation scores obtained on the training set are also important. For example, if evaluation scores on the training dataset are high, and on testing datasets are lower, it could potentially indicate over-fitting. So, in this paper, the idea is to show mean values of train/test scores with standard deviation. The large standard deviation between estimation accuracies achieved on the train and test datasets could potentially indicate that over-fitting occurred. On the other hand, the small standard deviation between estimation accuracies obtained on train and test datasets could indicate that the ML algorithm has stable, generalized, and robust estimation of targeted values. So, the standard deviation in this procedure is a key factor that could potentially indicate if over-fitting occurred.

The procedure for obtaining mean values of evaluation scores as well as standard deviation is very simple and consists of a couple of steps, i.e.,

- Train an ML algorithm on the training part of the dataset;
- Evaluate the ML algorithm on the training dataset using the previously mentioned evaluation metric methods;
- Evaluate the ML model on the testing dataset using previously mentioned evaluation metric methods;
- Calculate the mean value of train and test evaluation metrics scores using the formula

$$
\begin{aligned}
\overline{R^2} &= \frac{R^2_{\text{train}} + R^2_{\text{test}}}{2} \\
\overline{MAE} &= \frac{MAE_{\text{train}} + MAE_{\text{test}}}{2} \\
\overline{RMSE} &= \frac{RMSE_{\text{train}} + RMSE_{\text{test}}}{2};
\end{aligned}
\tag{3}
$$

- Calculate the standard deviation of train and test evaluation metric scores using formulas

$$R^2_{\text{STD}} = \sqrt{\frac{1}{N}\sum_{i=1}^{N}(R^2_i - \overline{R}^2)^2}$$

$$MAE_{\text{STD}} = \sqrt{\frac{1}{N}\sum_{i=1}^{N}(MAE_i - \overline{MAE})^2}$$

$$RMSE_{\text{STD}} = \sqrt{\frac{1}{N}\sum_{i=1}^{N}(RMSE_i - \overline{RMSE})^2}. \tag{4}$$

In the case of simple ML algorithm training using a classic split of the dataset, the $N$ in the previous equation is equal to 2.

## 3. Results

In this section, the results of the conducted investigation are presented. First, the results of the initial investigation are shown, i.e., the results of ML algorithms with default hyperparameters. The next step was to perform the 5-fold cross-validation on the training dataset with a randomized hyper-parameter search. Throughout these two investigations, the selection of ML algorithms was conducted, and only the best in terms of estimation accuracies and small standard deviation between train and test accuracies were used to build, train, and test the stacking ensemble model.

### 3.1. Results of Initial Investigation

The initial selection of previously mentioned ML algorithms is presented. The idea was to investigate the initial performance of the used ML algorithms and see their estimation performance with default parameters, i.e., a kind of "out of the box" approach. In Figure 9, the results of the estimation performance on the training and testing portion of the dataset are shown.

As seen in Figure 9, the algorithms used with default parameters have achieved extremely high values of $\overline{R}^2$ values, low $\overline{MAE}$ and $\overline{RMSE}$ values, and low standard deviation values. Generally, estimation accuracies ($\overline{R}^2$, $\overline{MAE}$, and $\overline{RMSE}$ values with extremely small standard deviation between train and test scores) are higher in the case of the black-box inverter compensation scheme ($d_{a,k-2}$, $d_{b,k-2}$, and $d_{c,k-2}$) than in the case of the black-box inverter model. However, two ML algorithms had poor estimation performance when compared with the others, and these are the ElasticNet and K-Nearest Neighbors in the case of the black-box inverter model output variables and MLP in the case of the black-box inverter compensation scheme output variables. The ElasticNet algorithm for $\overline{u}_{a,k-1}$, $\overline{u}_{b,k-1}$, and $\overline{u}_{c,k-1}$ achieved low $\overline{R}^2$ (0.65, 0.65, and 0.63) and high $\overline{MAE}$ (50.81, 50.07, and 51.25) and $\overline{RMSE}$ (67.7, 66.7, and 69.04) values. However, the standard deviation between estimation accuracies achieved on the training and testing parts of the dataset is small ($R^2_{STD} \approx 0.001$, $MAE_{STD} = 0.09$, and $RMSE_{STD} \approx 0.157$). Due to a small value of $\overline{R}^2$ and high values of $\overline{MAE}$ and $\overline{RMSE}$, these values are omitted from Figure 9. The K-Neighbors in the case of the black-box inverter model achieved low values of $\overline{R}^2$ for all three output variables ($\overline{u}_{a,k-1}$, $\overline{u}_{b,k-1}$, and $\overline{u}_{c,k-1}$) with the highest standard deviation values (indicated by the error bars in Figure 9), which indicates a high difference between $\overline{R}^2$ scores achieved on training and testing data, respectively. The achieved $\overline{MAE}$ and $\overline{RMSE}$ mean values in the case of K-Neighbors were applied on the black-box inverter model, where the mean values of these metrics are the highest when compared with the results achieved with other ML algorithms. The standard deviation values are the highest, which means there is a huge difference between $MAE$ and $RMSE$ values achieved on the training and testing

parts of the dataset. In the case of the MLP regressor, the estimation accuracies were high ($\approx 1$) in the case of the black-box inverter model output variables. However, in the case of the black-box inverter compensation scheme output variables, the estimation accuracies are very low, i.e., for the $d_{a,k-2}$, $d_{b,k-2}$, and $d_{c,k-2}$ output variables, the values of $\overline{R}^2$ are 0.06, 0.26, and 0.0398, respectively. Due to the small values of $\overline{R}^2$, they are not visible in Figure 9.

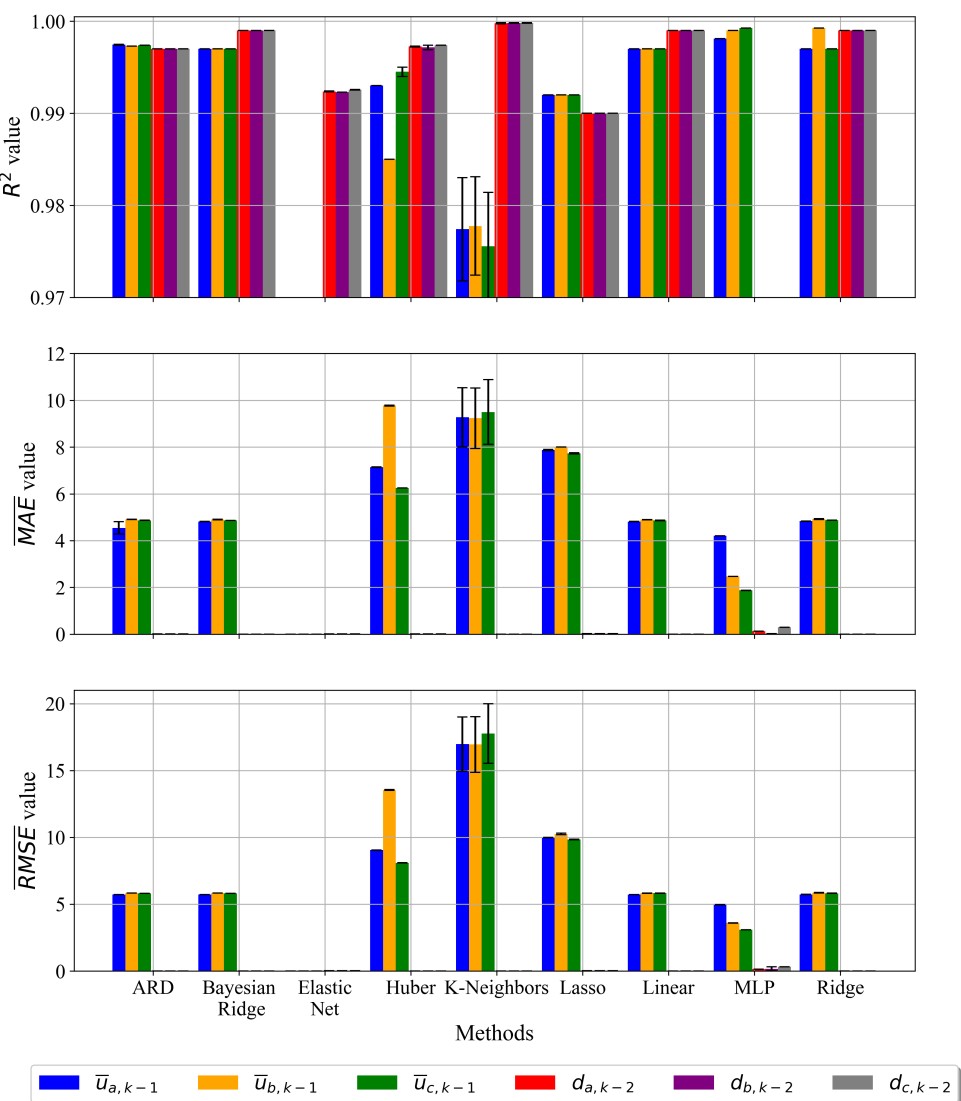

**Figure 9.** $\overline{R}^2$, $\overline{MAE}$, and $\overline{RMSE}$ values with standard deviation shown as error bars.

The default hyper-parameters of each ML algorithms that were used to obtain results are shown in Figure 9 are given in Table 12.

All the ML algorithms used in the initial investigation are used in a randomized hyper-parameter search with 5-fold cross-validation performed on the train part of the dataset to investigate if when using this method the estimation accuracies could be improved and the standard deviation lowered.

**Table 12.** Default hyper-parameters of ML algorithms used to obtain the results shown in Figure 9.

| ML algorithm | Hyper-parameters ($\overline{u}_{ak-1}$, $\overline{u}_{bk-1}$, $\overline{u}_{ck-1}$, $d_{ak-2}$, $d_{bk-2}$, $d_{ck-2}$) |
|---|---|
| ARD | number of iterations = 300, tolerance = $1 \times 10^{-3}$, alpha1 = $1 \times 10^{-6}$, alpha2 = $1 \times 10^{-6}$, lambda1 = $1 \times 10^{-6}$, lambda2 = $1 \times 10^{-6}$, threshold lambda = 10,000 |
| Bayesian ridge | number of iterations = 300, tolerance = $1 \times 10^{-3}$, alpha1 = $1 \times 10^{-6}$, alpha2 = $1 \times 10^{-6}$, lambda1 = $1 \times 10^{-6}$, lambda2 = $1 \times 10^{-6}$, alpha initial = None, lambda initial = None, |
| Huber | epsilon = 1.35, max_iter = 100, alpha = 0.0001, tolerance = $1 \times 10^{-5}$ |
| Elastic Net | alpha = 1.0, l1_ratio = 0.5, max_iter = 1000, tolerance = $1 \times 10^{-4}$ positive = False, selection = cyclic |
| K-Neighbors | n_neighbors = 5, weights = uniform, algorithm = auto, leaf_size = 30 p = 2, metric = minkowski, metric_params = None |
| Lasso | alpha = 1.0, fit_intercept = True, max_iter = 1000, tolerance = $1 \times 10^{-4}$ positive = False, random_state = None, selection = cyclic |
| Linear | fit_intercept = True, False |
| MLP | hidden_layer_sizes = (100), activation = relu, solver = adam, alpha = 0.0001, batch_size = auto, learning_rate = constant, learning_rate_init = 0.001, power_t = 0.5, max_iter = 200, shuffle = True, random_state = None, tolerance = $1 \times 10^{-4}$ |
| Ridge | alpha = 1.0, max_iter = None, tolerance = $1 \times 10^{-3}$, solver = auto, positive = False, random_state = None |

### 3.2. Randomized Hyper-Parameter Search with Cross-Validation

In this investigation, the hyper-parameters of each ML algorithm were randomly selected from a predefined range. The dataset, as in the previous case, was split into train and test datasets with the ratio of 70:30. Then, the training of each ML algorithm was performed on the training dataset using 5-fold cross-validation. After cross-validation, the estimation accuracies are calculated on the train and test datasets and used in termination criteria. If the values are above predefined estimation accuracies ($R^2 > 0.999$, $MAE < 7.0$, $RMSE < 10.0$), the model is trained again on the training dataset and evaluated on the test dataset. If not, then the parameters are randomly chosen and the cross-validation process is repeated until termination criteria are met. $\overline{R}^2$, $\overline{MAE}$, and $\overline{RMSE}$ values with standard deviation presented as error bars are shown in Figure 10.

From Figure 10, all ML algorithms achieved high estimation accuracies in both models (black-box inverter model and black-box inverter compensation scheme) with low standard deviation, except the K-Neighbors, so it was omitted from further investigations. It should be noted that extremely high estimation accuracies are achieved with all ML algorithms in the case of the black-box inverter compensation scheme output variables. The K-Neighbors algorithm showed the same problems, even though the randomized hyper-parameter search with 5-fold cross-validation was performed. The standard deviations were the same as in the case of the initial investigation with default parameters. The small but noticeable

standard deviations occurred in the case of MLP and Ridge regression for the case of the black-box inverter model output variables. The parameters for each ML algorithm achieved using the best estimation result shown in Figure 10 are given in Table 13.

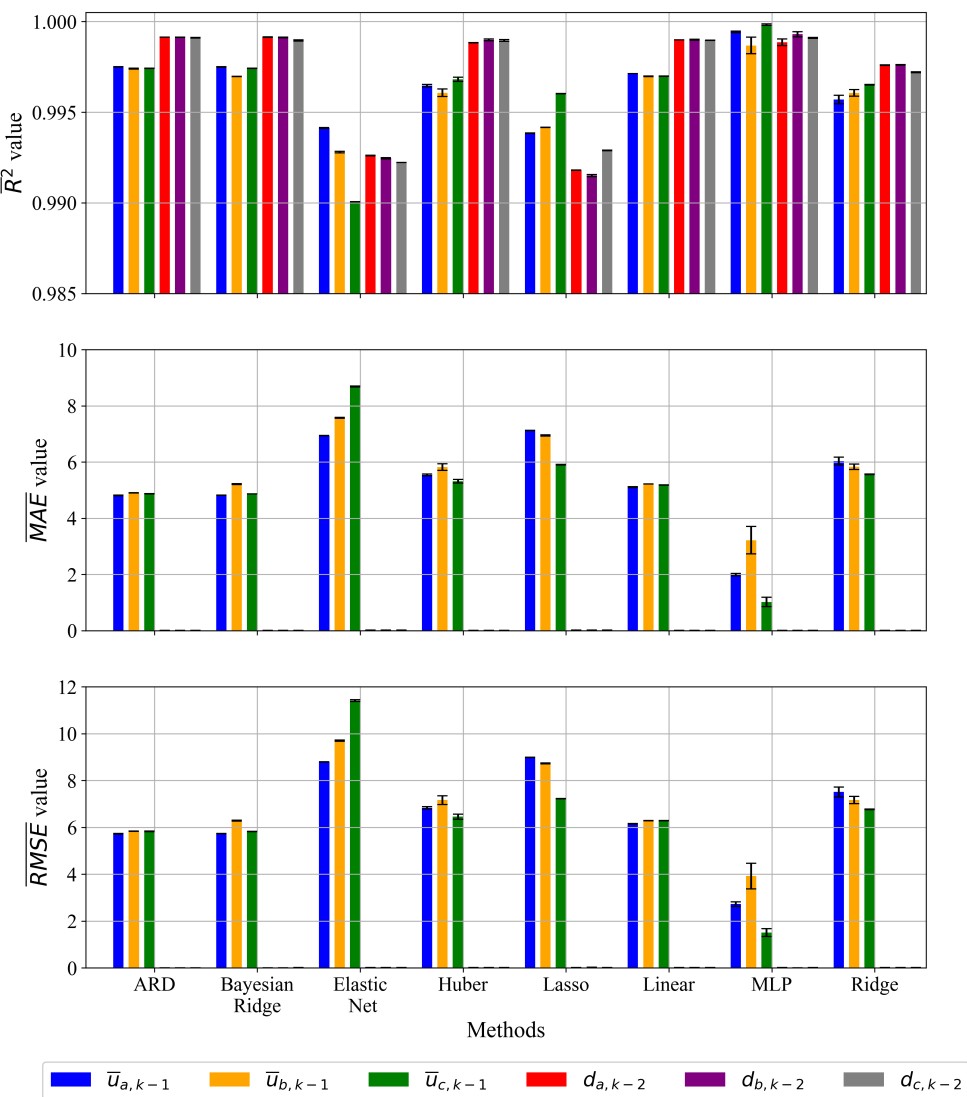

**Figure 10.** $\overline{R}^2$, $\overline{MAE}$, and $\overline{RMSE}$ values with standard deviation achieved with each ML algorithm using randomized hyper-parameter search with cross-validation.

From Table 13, it can be noticed that *tolerance* hyper-parameters for ML algorithms such as ARD, Huber, ElasticNet, and Lasso are very low ($10^{-21} - 10^{-6}$). The hyper-parameter value range was in some ML algorithms widened since the small ranges did not produce any significant benefit to the values of estimation accuracy.

The next and final step is to use all ML algorithms shown in Figure 10 and build the stacking ensemble to investigate if when using the ensemble method the estimation accuracies could somehow be improved.

**Table 13.** The hyper-parameters of each ML-algorithm in randomized hyper-parameter search with 5-fold cross-validation.

| ML Algorithm | Hyper-parameters ($\overline{u}_{ak-1}$, $\overline{u}_{bk-1}$, $\overline{u}_{ck-1}$, $d_{ak-2}$, $d_{bk-2}$, $d_{ck-2}$) |
|---|---|
| ARD | n_iter = 854, 286, 934, 555, 284, 354 <br> tolerance = $3.4624 \times 10^{-26}$, $9.8841 \times 10^{-26}$, $4.787 \times 10^{-26}$, $9.8719 \times 10^{-26}$, $7.764 \times 10^{-27}$, $6.02 \times 10^{-26}$ <br> alpha1 = 0.023148, 0.0648, 0.05917, 0.05919, 0.0947, 0.0237 <br> alpha2 = 0.02833, 0.00509, 0.01208, 0.09771, 0.05974, 0.058 <br> lambda1 = 0.0864, 0.04028, 0.03316, 0.03642, 0.0789, 0.076 <br> lambda2 = 0.06796, 0.00708, 0.08056, 0.03634, 0.02163, 0.0746 <br> threshold_lambda = 37,335, 21,754, 41,212, 59,922, 53,871 |
| Bayesian ridge | n_iter = 566, 784, 880, 830, 974, 733 <br> tolerance = 0.00028, 0.00049, 0.00033, 0.00055, 0.00099, 0.00059 <br> alpha1 = 0.0577, 0.036, 0.0045, 0.017, 0.062, 0.057 <br> alpha2 = 0.081, 0.0205, 0.0233, 0.025, 0.08, 0.098 <br> lambda1 = 0.052, 0.0558, 0.065, 0.037, 0.052, 0.09 <br> lambda2 = 0.0301, 0.006, 0.023, 0.065, 0.085, 0.066 <br> lambda_init = None, 6.329, 0.34, None, 3.368, None |
| Huber | epsilon = 1.703, 67.87, 72.22, 1.15, 1.631, 9.89 <br> max_iter = 80,657, 26,744, 14,533, 41,745, 24,066, 93,820 <br> alpha = 0.00027, 0.04, 0.0071, 0.00036, $2.623 \times 10^{-5}$, 0.00014 <br> tolerance = $3.77 \times 10^{-21}$, 0.0184, 0.07, $5.31 \times 10^{-21}$, $6.69 \times 10^{-21}$, $3.954 \times 10^{-21}$ |
| Elastic Net | alpha = 0.62, 0.048, 0.029, 0.94, 1.29, 0.51 <br> l1_ratio = 0.99, 0.93, 0.845, 0.36, 0.27, 0.98 <br> max_iter = 83,712, 11,323, 69,707, 17,535, 27,655, 41,066 <br> tolerance = $9.41 \times 10^{-6}$, $8.25 \times 10^{-6}$, $9.13 \times 10^{-6}$, $4.36 \times 10^{-6}$, $4.997 \times 10^{-6}$, $8.18 \times 10^{-6}$ <br> random_state = 33, 10, 9, 19, 20, 14 <br> selection = cyclic, random, cyclic, random, cyclic, random |
| Lasso | alpha = 0.74, 0.67, 0.32, 0.58, 0.76, 0.28 <br> max_iter = 2814, 9494, 1431, 3395, 2852, 2568 <br> tolerance = $5 \times 10^{-11}$, $8.57 \times 10^{-11}$, $9.44 \times 10^{-11}$, $5.45 \times 10^{-11}$, $5.328 \times 10^{-11}$, $5.27 \times 10^{-11}$ <br> random_state = None, None, None, 48, None, 7 <br> selection = cyclic, random, cyclic, cyclic, random, cyclic |
| Linear | fit_intercept = False, True, True, True, True, True |
| MLP | hidden_layer_size = (107, 139), (140, 76, 99, 64, 113), (139, 144, 128, 157), (111, 45, 121) (129, 71, 75), (152, 178, 110) <br> activation_function = logistic, relu, relu, relu, logistic, logistic <br> solver = adam, adam, adam, adam, adam, adam <br> alpha = 0.00019, 0.007, 0.005, 0.0078, 0.0028, 0.0059 <br> batch_size = 214, 246, 238, 277, 263, 220 <br> learning_rate = constant, invscaling, constant, adaptive, constant, constant <br> max_iter = 1153, 299, 1346, 338, 1169, 1967 <br> tolerance = $6.22 \times 10^{-5}$, $1.75 \times 10^{-5}$, $5.02 \times 10^{-5}$, $3.77 \times 10^{-5}$ $5.46 \times 10^{-5}$, $7.068 \times 10^{-5}$ <br> nIter_no_change = 956, 23, 1310, 193, 1140, 710 |
| Ridge | alpha = 682.580, 368.425, 254.265, 805.43, 453.45, 580.24 <br> fit_intercept = False, True, False, True , False, False <br> max_iter = 66,120, 78,658, 34,453, 25,145, 82,456, 75,458 <br> tolerance = $5.143 \times 10^{-5}$, $3.128 \times 10^{-7}$, $4.126 \times 10^{-4}$, $2.464 \times 10^{-3}$, $3.126 \times 10^{-4}$, $1.236 \times 10^{-8}$ <br> solve = sag, saga, cholesky, sag, saga, sag |

### 3.3. Ensemble Methods

In this section, the results obtained using a stacking ensemble are presented. To develop the stacking ensemble, all ML algorithms that were used in previous investigations are used as base estimators inside the staking ensemble. The *final_estimator*, which is used to combine the base estimator, is randomly selected from the base estimators list. Here, the analysis of each output variable was performed using the stacking ensemble with randomized hyper-parameter search with 5-fold cross-validation on the train part of the dataset and final evaluation on the test part of the dataset. The results ($\overline{R^2}$, $\overline{MAE}$, and $\overline{RMSE}$ with standard deviation) achieved in the estimation of variables in the black-box inverter model and black-box inverter compensation scheme are shown in Figure 11.

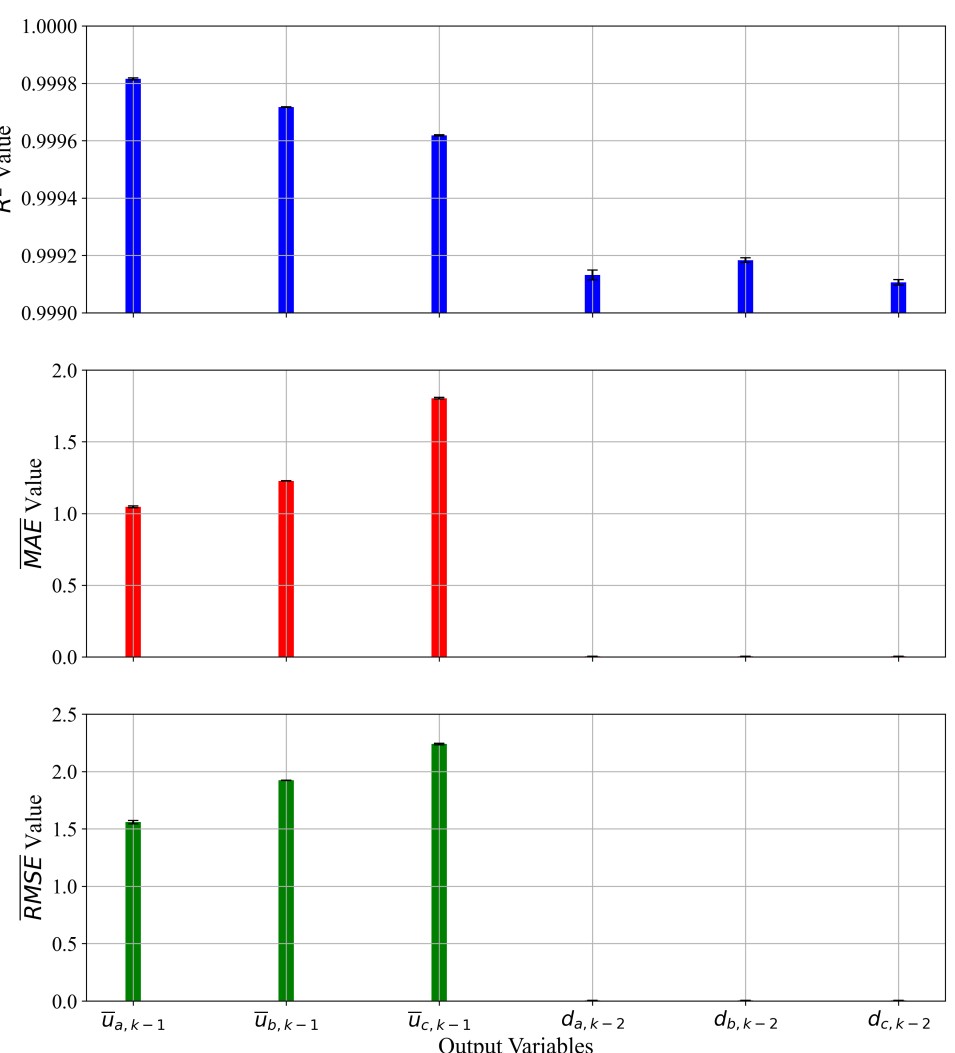

**Figure 11.** $\overline{R^2}$, $\overline{MAE}$, and $\overline{RMSE}$ values for each output variable with standard deviation values.

From Figure 11, it can be seen that estimation accuracies in the case of the black-box inverter compensation scheme output variables are almost perfect, i.e., the values of $\overline{R^2}$ are almost 1.0 (<0.9997) with extremely low $\overline{MAE}$ and $\overline{RMSE}$ values ($\approx$0.001). The standard deviation values in the case of the black-box inverter compensation scheme output variables are virtually nonexistent, which means that the difference between estimation accuracies achieved on the train and test part of the dataset is almost the same. The implementation of the stacking ensemble with randomized hyper-parameter search and classic train/test split of the dataset also showed improvement. $\overline{R^2}$ values are also near 1.0 ($\approx$0.99) while values of $\overline{MAE}$ and $\overline{RMSE}$ are lowered when compared with the previous investigation,

where the highest $\overline{MAE}$ and $\overline{RMSE}$ values were above 8 and 11 and, here, the highest $\overline{MAE}$ value and $\overline{RMSE}$ values are slightly above 1.5 and 2.0 (in case of $\overline{u}_{c,k-1}$). These results were achieved with the stacking ensemble where the *final_estimator* is the MLP that had the same hyper-parameters as the base MLP estimator for each output variable. The parameters of each ML algorithm used in the stacking ensemble to obtain the result presented in Figure 11 are shown in Table 14.

**Table 14.** The hyper-parameters of each ML-algorithm used in the stacking ensemble with randomized hyper-parameter search and 5-fold cross-validation.

| ML algorithm | Hyper-parameters ($\overline{u}_{ak-1}, \overline{u}_{bk-1}, \overline{u}_{ck-1}, d_{ak-2}, d_{bk-2}, d_{ck-2}$) |
|---|---|
| ARD | num_iter = 804, 957, 808, 375, 828, 813<br>tol = $3.34 \times 10^{-26}$, $4.871 \times 10^{-27}$, $1.375 \times 10^{-26}$,<br>$5.785 \times 10^{-26}$ $6.439 \times 10^{-27}$ $8.04 \times 10^{-26}$<br>alpha_1 = 0.064, 0.0689, 0.0772, 0.0707, 0.0674, 0.03,<br>alpha_2 = 0.0868, 0.0819, 0.0603, 0.0432, 0.0936, 0.063<br>lambda_1 = 0.0351, 0.0655, 0.0839, 0, 073, 0.068, 0.007<br>lambda_2 = 0.0914, 0.0323, 0.0983, 0, 083, 0.0904, 0.08,<br>compute_score = True, True, False, True, True, True<br>threshold_lambda = 21,256, 89,378, 43,190, 60,432, 69,129, 26,796 |
| Bayesian ridge | num_iter = 997, 837, 948, 504, 777, 656<br>tol = 0.0003, 0.00063, 0.000707, 0.000546, 0.00078, 0.0005<br>alpha_1 = 0.0155, 0.036, 0.062, 0.0051, 0.0094, 0.058<br>alpha_2 = 0.0437, 0.055, 0.055, 0.065, 0.027, 0.022<br>lambda_1 = 0.0334, 0.019, 0.0103, 0.0576, 0.0996<br>lambda_2 = 0.0991, 0.019, 0.031, 0.0135, 0.0482, 0.076<br>lambda_init = 1.7039, 1.815, 5.966, None, 4.96, None<br>compute_score = False, False, False, True, False, False<br>fit_intercept = True, False, False, False, True, False |
| Huber | epsilon = 9.68, 5.699, 7.41, 6.08, 1.36, 4.79<br>max_iter = 67884, 90043, 42378, 19418, 32229, 68369,<br>alpha = 0.000261, $4.73 \times 10^{-6}$, 0.000617, 0.00067, 0.00037, $2.7 \times 10^{-6}$<br>fit_intercept = True, True, True, True, True, True<br>tol = $7.33 \times 10^{-21}$, $7.206 \times 10^{-21}$, $9.34 \times 10^{-21}$, $3.52 \times 10^{-21}$<br>$8.135 \times 10^{-21}$, $1.23 \times 10^{-21}$ |
| Elastic Net | alpha = −9.79, 2.09, −9.98, −9.264, −2.35, −1.13<br>l1_ratio = 0.245, 0.143, 0.37, 0.924, 0.27, 0.9<br>fit_intercept = False, False, False, False, False, False<br>max_iter = 32841, 41891, 35791, 54745, 54209, 84589<br>tol = $5.678 \times 10^{-7}$, $3.678 \times 10^{-6}$, $8.99 \times 10^{-6}$, $9.93 \times 10^{-6}$<br>$3.61 \times 10^{-6}$, $8.86 \times 10^{-6}$<br>random_state = 28, 25, 31, 42, 3, 13<br>selection = random, random, random, random, random, cyclic |
| Lasso | alpha = 5.38, 7.606, 0.334, 2.01, 1.026, 5.52<br>fit_intercept = True, True, False, True, False, True<br>max_iter = 8423, 9285, 4220, 2606, 6958, 8770<br>tol = $6.75 \times 19^{-10}$, $9.92 \times 10^{-11}$, $9.92 \times 10^{-11}$, $3.027 \times 10^{-11}$<br>$4.55 \times 10^{-11}$, $9.19 \times 10^{-11}$<br>random_state = None, None, 27, 45, 16, None,<br>selection = cyclic, cyclic, cyclic, cyclic, random, random |

**Table 14.** *Cont.*

| Linear | fit_intercept = True, True, False, True, False, False |
|---|---|
| MLP | hid_layer_size = (70, 125, 157, 187), (18, 118, 199), <br> (162, 191, 172), (130, 176, 53, 40, 111), <br> (107, 26, 116), (174, 170, 88, 122, 111) <br> activation = tanh, relu, relu, identity, logistic, identity <br> solver = lbfgs, adam, adam, adam, lbfgs, adam <br> alpha = 0.0066, 0.00201, 0.0039, 0.00158, 0.0063, 0.0072 <br> batch_size = 264, 247, 285, 222, 290, 287 <br> learning_rate = constant, constant, constant, adaptive, invscaling, invscaling <br> max_iter = 1108, 447, 1942, 1942, 1436, 1755 <br> tol = $3.19 \times 10^{-6}$, $6.427 \times 10^{-5}$, $2.023 \times 10^{-5}$, <br> $6.0536 \times 10^{-5}$, $8.39 \times 10^{-5}$ $6.75 \times 10^{-5}$ <br> n_iter_no_change = 195, 218, 282, 959, 768, 494 |
| Ridge | alpha = 813.31, 149.53, 829.864, 897.206, 590.799, 846.2 <br> fit_intercept = False, True, False, True, False, True <br> max_iter = 25,174, 60,066, 58,733, 45,444, 84,797, 60,136 <br> tol = $1.08 \times 10^{-8}$, 0.000618, 0.00095, 0.00039 <br> 0.000669, 0.00078 <br> solver = auto, lsqr, auto, lsqr, svd, lsqr |

## 4. Discussion

The initial investigation using ML algorithms with default hyper-parameters with the classic split of the dataset on training and testing data with a ratio of 70:30 showed that the majority of ML algorithms have achieved high estimation accuracies. The algorithms achieved high $\overline{R}^2$ and low $\overline{MAE}$ and $\overline{RMSE}$ values. This is valid for all ML algorithms except the ElasticNet, K-Neighbors, and MLP. In the case of ElasticNet, K-Neighbors' estimation performance was poor in the case of the black-box inverter model output variables. In the case of the MLP, the estimation performance was poor in the case of the black-box inverter compensation scheme output variables. The standard deviation between training and testing estimation accuracies $R^2_{STD}$, $MAE_{STD}$, and $RMSE_{STD}$ were low for the majority of ML-trained algorithms, which indicates that over-fitting did not occur, except for K-Neighbors in the case of the estimation of the black-box inverter model output variables. The standard deviation values for that case indicate that differences between estimation accuracies on the training and testing datasets are large, which can be noticed with the error bars shown in Figure 9.

All the algorithms used in the initial investigation are used in a randomized hyper-parameter search with 5-fold cross-validation performed on the training dataset to investigate if estimation accuracies could be improved and standard deviation lowered. The investigation showed that almost all the ML algorithms achieved higher estimation accuracies (higher than the initial investigation), except for the K-Neighbors, which showed similar behavior as in the initial investigation with the default parameter. Although the estimation performance was somewhat improved, the standard deviation, i.e., the difference between the estimation accuracy achieved on the train and test dataset, was large for both models. So, this algorithm was omitted from further investigation using a stacking ensemble. In the initial investigation, the ElasticNet achieved lower estimation accuracies in the case of the black-box inverter model output variables, however, in the case of the randomized hyper-parameter search with 5-fold cross-validation, the estimation accuracies were improved. In the case of the MLP regression algorithm in the initial investigation, the estimation accuracies were extremely low ($\overline{R}^2 \approx 0.06$) for the case of the black-box inverter compensation scheme output variables, however, in the randomized hyper-parameter search with 5-fold cross-validation, the estimation accuracies were also improved ($\overline{R}^2 \approx 0.997$). The standard deviation between estimation accuracy values is virtually nonexistent for the majority of ML algorithms, however, the standard deviation can be noticed in the case of the MLP regressor for the black-box inverter model out-

put variables. So, the results of this investigation show that all ML algorithms will be used to build the stacking ensemble, except for K-Neighbors, which was omitted from further investigation.

As already stated in the final investigation, the stacking ensemble was used to investigate if estimation accuracy values could be improved with the ensemble method. For this ensemble, eight different ML algorithms were used, and they are ARD, Bayesian Ridge, Elastic Net, Huber, Lasso, Linear, MLP, and Ridge. Here, the stacking ensemble was investigated using a randomized hyper-parameter search of each ML algorithm and the application of 5-fold cross-validation on the training part of the dataset, while the test dataset was used for final evaluation.

The investigation using a stacking ensemble showed that combining all eight ML algorithms improved estimation accuracy values when compared with the previous investigation. $\overline{R}^2$ values for all six output variables (both models) are almost equal to 1.0. $\overline{MAE}$ and $\overline{RMSE}$ values were lowered in the case of the black-box inverter model output variables when compared with all previous investigations. The highest $\overline{MAE}$ and $\overline{RMSE}$ values are in the case of $\overline{u}_{c,k-1}$, where $\overline{MAE} = 1.8$ and $\overline{RMSE} = 2.24$, respectively. However, one key disadvantage is that training time of the stacking ensemble is much higher than training an individual ML algorithm.

In Table 15, the discussion is summarized with improvements obtained at each stage of the methodology development.

**Table 15.** Summarized discussion with improvements obtained at each stage of methodology development in the case of the black-box inverter and inverter compensation scheme model.

| Stage | Description |
|---|---|
| Initial investigation | Advantages:<br>-good estimation accuracies $(\overline{R}^2, \overline{MAE}, \overline{RMSE})$,<br>-accuracies of duty cycles better than<br>in the case of mean phase voltages<br>Disadvantages:<br>-large STD between estimation accuracies<br>achieved on the train and test dataset<br>in case of KNN |
| Random grid search with 5-Fold Cross-Validation | Advantages:<br>-improved average estimation accuracy,<br>-mean phase voltages and duty cycle accuracies improved,<br>-STD between accuracies on the train and test dataset lowered<br>Disadvantages:<br>-KNN omitted from further investigation due to the same STD<br>as in previous case<br>-ElasticNet and MLP showed slight increase<br>in $R_{STD}^2$, $MAE_{STD}$, and $RMSE_{STD}$ values |
| Ensemble Method | Advantages:<br>-estimation accuracies are optimal since<br>algorithm combines estimations of 8 basic estimators<br>Disadvantages:<br>-training time longer than training individual ML algorithm |

## 5. Conclusions

In this paper, various ML methods were used to estimate mean phase voltages at $k - 1$ of the black-box inverter model and duty cycles $k - 2$ of the black-box inverter compensation scheme, respectively. The initial investigation showed that the majority of used ML methods had high accuracy with default hyper-parameters. The investigation with randomized hyper-parameter search and 5-fold cross-validation had a slight improvement in estimation accuracies. Based on the conducted investigation, the following conclusions are:

- The initial investigation showed that with the original dataset and ML algorithms with default parameters good estimation accuracies could be achieved, so it was not necessary to perform classic scaling and normalization techniques on the dataset.
- The randomized hyper-parameter search with 5-fold cross-validation on the training dataset showed that estimation accuracy values were improved for the majority of ML algorithms with the exception of K-Neighbors, which showed the same behavior as in the previous investigation.
- The final investigation with stacking ensemble with randomized hyper-parameter search used for each ML algorithm and with 5-fold cross-validation showed improved estimation performance when compared with the previous case.

Using the artificial intelligence approach to estimate the parameters of power electronics circuits, it is possible to obtain high-quality predictions of the target variables of the inverter itself. The results of this research show the high accuracy of the target values, in this case, the black-box inverter model and the black-box inverter compensation model, which was also the goal of this research. The model follows the duty cycle estimation with great reliability, and the given model has a high accuracy for the mean phase voltage for $k - 1$, which contributes to the importance of integrating the AI algorithms themselves into power electronics circuits. Estimation of the mean phase voltage has a crucial role for devices of a sensitive nature in an environment that requires high accuracy of the power supply for efficient operation.

Regarding future research plans and contributions in this branch of science, several actions will be taken. It will be necessary to process the effect of heat on power electronics circuits because the effect of temperature on the power electronics circuit itself can also affect its properties. By increasing or decreasing the temperature, different triggering characteristics of the transistor can appear, which can greatly affect the waveform conversion of the circuit. In addition to the above, it will be necessary to make appropriate heat regulations for maintaining a constant temperature of the inverter itself (cooling or heating), which can affect the efficiency of the circuit itself. After performing the previously defined actions, it will be necessary to create a realistic model of the system itself with the associated DC power supply and inverter connected to the electronic drive and, after that, it will be possible to implement the obtained artificial intelligence model for estimating the target variables. With the successful execution of these actions, the scientific research on this topic will be completed.

**Author Contributions:** Conceptualization, N.A., I.L. and Z.C.; methodology, N.A., M.G. and Z.C.; software, N.A., I.L. and Z.C.; validation, I.L., M.G. and Z.C.; formal analysis, N.A., M.G. and I.L.; investigation, N.A. and I.L.; resources, Z.C.; data curation, I.L., M.G.; writing—original draft preparation, N.A., I.L. and Z.C.; writing—review and editing: N.A., M.G. and I.L.; visualization, I.L. and Z.C.; supervision, Z.C.; project administration, Z.C.; funding acquisition, Z.C. All authors have read and agreed to the published version of the manuscript.

**Funding:** This research has been (partly) supported by the CEEPUS network CIII-HR-0108, European Regional Development Fund under the grant KK.01.1.1.01.0009 (DATACROSS), project CEKOM under the grant KK.01.2.2.03.0004, Erasmus+ project WICT under the grant 2021-1-HR01-KA220-HED-000031177 and the University of Rijeka scientific grant uniri-tehnic-18-275-1447.

**Institutional Review Board Statement:** Not applicable.

**Informed Consent Statement:** Not applicable.

**Data Availability Statement:** All the Python scripts developed and used in this research is made availiable at: https://github.com/nandelic2022/InverterML/tree/main (accessed on: 20 July 2022).

**Conflicts of Interest:** The authors declare no conflict of interest.

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
