# Peer review of "Mean Phase Voltages and Duty Cycles Estimation of a Three-Phase Inverter in a Drive System Using Machine Learning Algorithms"

_electronics, doi:10.3390/electronics11162623_

Round 1

Reviewer 1 Report

This manuscript describes ML techniques (standalone and ensemble) ap­plied to estimate the mean phase voltages and duty cycles of the black-box inverter model and black box compensation scheme using a "Three-Phase IGBT Two-Level Inverter for Electrical Drives" dataset.

Although I have not found other works that use the same dataset, the authors must justify the contribution of the submitted manuscript. The contribution has not been made clear throughout the text. Is it just a ML ensemble in a standard date set? Is it possible to compare the results of other similar works?

The initial data evaluation is quite interesting despite not being used for preprocessing.

Minors:

- Line 31: "60 Hz. However..."

- Some occurrences of "black box" instead and "black-box".

-Section 4 must have a table that summarizes the discussion. Please, presents the improvements obtained at each stage of the methodology development: ML without parameter optimization, ML with parameter optimization, and ensemble.

-Is Section 2.7.1 necessary?

-Section 3.1 summarizes the main results of each ML in a table considering black-box inverter

black-box compensation

-Summary table with the main features of the ML techniques presented, i. ex. supervisioned/regression, generalization, etc.

Author Response

The authors want to thank the reviewer for his time, effort and constructive suggestions that have greatly improved the quality of this manuscript. The authors of this manuscript hope that the changes made in this manuscript will provide a suitable scientific contribution.

This manuscript describes ML techniques (standalone and ensemble) ap­plied to estimate the mean phase voltages and duty cycles of the black-box inverter model and black box compensation scheme using a "Three-Phase IGBT Two-Level Inverter for Electrical Drives" dataset.

  1. Although I have not found other works that use the same dataset, the authors must justify the contribution of the submitted manuscript. The contribution has not been made clear throughout the text. Is it just a ML ensemble in a standard date set? Is it possible to compare the results of other similar works?

The authors have made an effort during the manuscript development to compare obtained results with results obtained by other researchers. To the best of our knowledge, we did not find any research paper where ML algorithms were used on this dataset. Regarding the contribution in the last 3 paragraphs of the Introduction section, the authors have made a clear and explicit contribution as well as a hypothesis definition.

Citing from modified version of the manuscript:

In this paper, the idea is to investigate the possibility of utilizing a complex ML ensemble system (stacking ensemble) to estimate the mean phase voltages and/or duty cycles of three-phase IGBT two-level inverter for electrical drives. The ML ensemble system is a technique that combines basic ML algorithms to produce one optimal estimation model. So the idea is to develop an optimal estimation model using an ML ensemble system that can estimate mean phase voltages and/or duty cycles. Generally, two different models were considered:

  • black-box inverter model and
  • black-box inverter compensation scheme,

where the term “black-box” refers to the utilization of complex ML algorithms. In the black-box inverter model, the goal is to obtain ML algorithms that could estimate the mean phase voltages of the inverter with high accuracy. In a black-box inverter compensation scheme, the goal is to obtain ML algorithms that could estimate the duty cycles of the inverter with high accuracy.

To build this complex ensemble system selection of basic ML algorithms is required in specific steps. The first step is to investigate which one of the available ML algorithms can achieve good estimation accuracy with default parameters which can be described as an "out of the box" approach. The second step is to investigate the random hyper-parameter search with cross-validation to see if estimation accuracies could be improved. Finally, those ML algorithms that achieved the highest accuracies are randomly selected in the training process of ensemble methods to see which combination of estimators achieves the highest estimation accuracy.

To summarize, the hypothesis of this research are:

  • is it possible to achieve high estimation accuracies of black-box inverter models and black-box inverter compensation scheme targeted variables , using different ML algorithms with default parameters,
  • is it possible to improve the estimation accuracies of black-box inverter models and black-box inverter compensation scheme targeted variables, with random hyper-parameter grid search with 5-fold cross-validation applied on ML algorithms used in the previous step, and
  • is it possible to develop the stacking ensemble (using ML algorithms that achieved the highest estimation accuracies in the previous step) and on that stacking ensemble apply the random hyper-parameter grid search with 5-fold cross-validation to achieve high estimation accuracy with improved generalization and robustness of targeted variables in black-box inverter model and black-box inverter compensation scheme.
  1. The initial data evaluation is quite interesting despite not being used for preprocessing.

This comment

The two models analyzed in this manuscript differ  based on the chosen input and output (target) variables. The main reason for selecting this combination of input and output variables in each model was based on Pearson’s correlation analysis. This means that input variables with high correlation to the output variables were selected for each model. It should be noted that the authors in paper [1] only proposed similar set of input and output variables for each model. However, authors in paper [1] suggested that DC-link voltage at k and k-1 are part of input variables in case of black-box inverter model. They have also suggested that DC-link voltage at k-2 and k-3 are part of input variables in case of black-box inverter compensation scheme. These variables were shown in Figures 4 and 5 from which it can be seen that these variables have lowest correlation to the target output variables in each model. It should also be noted that speed [rev/min] at k is one of the variables in the dataset. However, this variable does not have any correlation (= 0) with desired target variables in each model respectively so it was not used in further investigation.

[1]  Stender, M.; Wallscheid, O.; Böcker, J. Data Set Description: Three-Phase IGBT Two-Level Inverter for Electrical Drives.

The authors have created a new paragraph at the end of subsubsection entitled Dataset statistical analysis in which they have given previously written explanation.

Citing paragraph from modified version of the manuscript:

It should be noted that these black-box inverter models were created for two reasons. The authors in [38] have suggested these models (combination of input and output variables). This suggestion of data configuration for both models was tested and verified with conducted Pearson’s correlation investigation which showed an excellent correlation between input and output variables and can be seen for each model in Figures 4, and 5, respectively. However, the investigation showed that each black-box inverter model has two variables with the lowest correlation to the target variables. As already stated, in black-box inverter model the DC-link voltage at k (udc,k) and DC-link voltage at k − 1 (udc,k−1) have lowest correlation (≈ 0) with target variables (ua,k−1, ub,k−1, and uc,k−1). In case of black-box inverter compensation scheme the DC-link voltage at k − 2 and k − 3 do not have any correlation with target variables (da,k−2, db,k−2, and dc,k−2). The DC-link voltages at k, k − 1, k − 2 and k − 3 were suggested by the authors in [38] so the results of the correlation analysis for these variables was left in Figures 4 and 5. The dataset also contains the speed at k. However, this variable does not have any correlation (≈ 0) with the target variables in both models.

The authors of this paper were initially considering the implementation of data preprocessing techniques. However, the investigations conducted with the original dataset (without preprocessing) showed outstanding estimation accuracies of targeted variables. So, the implementation of preprocessing techniques was omitted from further investigations in this paper. This was also mentioned in the manuscript. Citing from the original manuscript (lines 171 - 182, page 6): “Initially, the application of scaling and normalization techniques on the dataset was considered however, the high accuracy was achieved with ML algorithms on the original dataset so the application of data preprocessing methods was omitted from further investigations.

Minors:

  1. - Line 31: "60 Hz. However..."

The error was corrected.

The original version of the sentence (lines 29-31): Common types of power inverters produce square waves or quasi-square waves [6], where the output frequency of AC is usually 50 or 60 Hz however, in power inverter designs used for motor driving the variable frequency results in variable speed control.

The modified version of the sentence (lines 29-31): Common types of power inverters produce square waves or quasi-square waves [6], where the output frequency of AC is usually 50 or 60 Hz. However, in power inverter designs used for motor driving the variable frequency results in variable speed control.

  1. - Some occurrences of "black box" instead and "black-box".

The occurrences of “black box” were replaced with “black-box”.

The part of sentence at line 6 (original manuscript, the “-” missing between words “black” and “box”): “... inverter model and black box inverter compensation…”

The part of the sentence at line 6 (modified manuscript, the “-” added between “black” and “box”): “...inverter model and black-box inverter compensation…”

The Table 1 title after line 151 (original manuscript version, the “-” missing between words “black” and “box”): “Table 1. Statistical description of the black box inverter model (The k refers to sampling step).

The Table 1 title after line 151 (modified manuscript version, the “-” added between words “black” and “box”): “Table 1. Statistical description of the black-box inverter model (The k referees to sampling step).”

The Table 2 title at beginning of the page 6 (original manuscript version, the “-” missing between words “black” and “box”): “Table 2. Statistical description of the black box inverter compensation scheme. (The k refers to sampling step).

The Table 2 title at beginning of the page 6 (modified manuscript version, the “-” added between words “black” and “box”): “Table 2. Statistical description of the black-box  inverter compensation scheme. (The k refers to sampling step).

The Figure 4 title before the line 197 (original manuscript version, the “-” missing between words “black” and “box”): “Figure 4. Pearson’s correlation heat-map for black box inverter model.”

The Figure 4 title before the line 197 (modified manuscript version, the “-” added between words “black” and “box”): “Figure 4. Pearson’s correlation heat-map for black-box inverter model.”

The Figure 5 title at the beginning of page 8 (original manuscript version, the “-” missing between words “black” and “box”): “Figure 5. Pearson’s correlation heat-map for black box inverter compensation scheme.”

The Figure 5 title at the beginning of page 8 (modified manuscript version, the “-” added between words “black” and “box”): “Figure 5. Pearson’s correlation heat-map for black-box inverter compensation scheme.”

The line 617 (original manuscript version, the “-” missing between words “black” and “box”): “...k-1 of black box inverter model and duty cycles k-2 of black box compensation scheme,...”.

The line 617 (modified manuscript version, the “-” added between words “black” and “box”): “...k-1 of black-box inverter model and duty cycles k-2 of black-box compensation scheme,...”.

  1. -Section 4 must have a table that summarizes the discussion. Please, presents the improvements obtained at each stage of the methodology development: ML without parameter optimization, ML with parameter optimization, and ensemble.

At the end of section 4 entitled “Discussion” (page 27) the table that summarizes the discussion was added.

  1. -Is Section 2.7.1 necessary?

The authors agree that subsection 2.7.1 is not necessary so the entire 2.7 section was modified. First, it was renamed from Evaluation Metric and Methodology to only Evaluation Methodology. In the first paragraph of the Evaluation Methodology, the used evaluation metrics were mentioned.

Citing from the modified version of section 2.7 (Evaluation Methodology): “To evaluate the estimation accuracies of all ML algorithms in this paper three types of evaluation metrics were utilized i.e.: Coefficient of determination ($R^2$)[51], mean absolute error ($MAE$)[52], and root mean squared error ($RMSE$)[53].

  1. -Section 3.1 summarizes the main results of each ML in a table considering black-box inverter,black-box compensation

The Section 3.1. summarizes the results obtained from initial investigation i.e. each ML algorithm is trained with default parameter (all default parameters are given in scikit-learn webpage: https://scikit-learn.org/stable/modules/classes.html)

  1. -Summary table with the main features of the ML techniques presented, i. ex. supervisioned/regression, generalization, etc.

Summary table is sufficient since all ML algorithms were trained using supervised learning method with dataset initially divided in 70:30. It should be noted that 70% of dataset (training dataset) was used in 5-fold cross validation.

Reviewer 2 Report

This model of this paper needs to be clarified.

Please check are there any typos

Author Response

The authors want to thank the reviewer for his time, effort and constructive suggestions that have greatly improved the quality of this manuscript. The authors of this manuscript hope that the changes made in this manuscript will provide a suitable scientific contribution.

  1. This model of this paper needs to be clarified.

The authors have described in detail the procedure of how the dataset was developed. For detailed information check the subsection 2.1 of the modified manuscript version.

  1. Please check are there any typos

The authors have made an effort to check all the typos in the manuscript. To the best of our knowledge, all typos in the manuscript were corrected.

Reviewer 3 Report

In this work, ML analysis (learning) of inverter parameters is performed using an artificial open source database. In this thesis we use the journal "Sensors". Please refer to the link between the thesis and Sensors. The results of the article show that the task was performed with excellent statistical results. However, the article is flawed on several points. 1. What hardware is included in the database used. The material identified as number 36 in the bibliography is not available is not known and therefore not acceptable in this form. This is a technical research and technical journal and we are not applying algorithms for their own sake, we expect technical answers. Unfortunately this is not found in this work, which makes it difficult to evaluate the article. The description of the algorithms used cannot be followed, although this would be a crucial part of an otherwise 29-page article, which could be described as voluminous. I consider it a serious error to omit the various figures and numerical values by omitting the original source code. In my opinion, the work can be evaluated by bringing the ML algorithms to open platform, which can only be achieved by uploading this source code to GitHub. In this case, the reader can follow the algorithms
The thesis is not free of bugs and some serious errors (figures are illegible, it writes Hubber instead of Huber.

Author Response

The authors want to thank the reviewer for his time, effort and constructive suggestions that have greatly improved the quality of this manuscript. The authors of this manuscript hope that the changes made in this manuscript will provide a suitable scientific contribution.

  1. In this thesis we use the journal "Sensors". Please refer to the link between the thesis and Sensors.

The authors do want to apologies to the reviewer for the mistake made in the manuscript. On the header of each manuscript page from page 2 up to page 29 it states Version July 20, 2022 submitted to Sensors. However, the authors were considering to submit the manuscript to Sensors or Electronics but at the end the manuscript was sent to Electronics. Before initially submitting the manuscript the authors did not check the headers of the manuscript and so the error "submitted to Sensors" remained instead of "submitted to Electronics".

  1. What hardware is included in the database used.

The authors have described in detail the procedure of how the dataset was developed. They have also provided detailed information about entire system and the equipment used for measurement. To clarify all figures in this description were not copied from the other literature. For detailed information check the subsection 2.1 of the modified manuscript version.

  1. The material identified as number 36 in the bibliography is not available is not known and therefore not acceptable in this form.

The material identified as number 36 in the original version of the manuscript is the paper describing the dataset used in this research. In that paper there is a direct link to the dataset which can then be downloaded. The authors have corrected the reference (now under the number 37) and now this reference is the direct hyperlink to the dataset.

  1. This is a technical research and technical journal and we are not applying algorithms for their own sake, we expect technical answers. Unfortunately this is not found in this work, which makes it difficult to evaluate the article.

The algorithms have not been applied for their own sake. The idea was to see if stacking ensemble method could be used to make optimal estimation of each mean phase voltages and duty cycle variables. If this stacking ensemble can be obtained with cheap equipment (laptop) than it could be potentially implemented on real inverter. In this work, the idea was not to achieve an extremely high estimation accuracy and show it only on the training data set. In this paper, the results on the training and testing data sets (mean value of R^2, MAE, RMSE and standard deviation) are explicitly presented. To achieve this goal the authors had to initially choose ML algorithms and test their performance with default parameters. Since the algorithms showed good estimation performance without data preprocessing the results were shown in the paper. Then the randomized hyper-parpameter search was performed with 5-fold cross-validation to see if the estimation accuracies could be improved. Finally those ML algorithms that passed previous stage in terms of good estimation accuracies and low standard deviation of these values were selected as base estimators of stacking ensemble which again was trained with random hyper-parameter grid search with 5-fold cross-validation performed on each based estimator.

Citing from modified version of the manuscript (lines 74 to 108):

In this paper, the idea is to investigate the possibility of utilizing a complex ML ensemble system (stacking ensemble) to estimate the mean phase voltages and duty cycles of three-phase IGBT two-level inverter for electrical drives. The ML ensemble system is a technique that combines basic ML algorithms to produce one optimal estimation model. So the idea is to develop an optimal estimation model using an ML ensemble system that can estimate mean phase voltages and duty cycles. Generally, two different models were considered:

  • black-box inverter model and
  • black-box inverter compensation scheme,

where the term “black-box” refers to the utilization of complex ML algorithms. In the black box inverter model, the goal is to obtain ML algorithms that could estimate the mean phase voltages of the inverter with high accuracy. In a black-box inverter compensation scheme, the goal is to obtain ML algorithms that could estimate the duty cycles of the inverter with high accuracy. To build this complex ensemble system selection of basic ML algorithms is required in specific steps. The first step is to investigate which one of the available ML algorithms can achieve good estimation accuracy with default parameters which can be described as an "out of the box" approach. The second step is to investigate the randomized hyper-parameter search with cross-validation to see if estimation accuracies could be improved. Finally, those ML algorithms that achieved the highest accuracies are randomly selected in the training process of ensemble methods to see which combination of estimators achieves the highest estimation accuracy. To summarize, the hypothesis of this research are:

  • is it possible to achieve high estimation accuracies of black-box inverter models and black-box inverter compensation scheme targeted variables1 , using different ML  algorithms with default parameters,
  • is it possible to improve the estimation accuracies of black-box inverter models and black-box inverter compensation scheme targeted variables1, with randomized hyper-  parameter search with 5-fold cross-validation applied on ML algorithms used in the  previous step, and
  • is it possible to develop the stacking ensemble (using ML algorithms that achieved the highest estimation accuracies in the previous step) and on that stacking ensemble  apply the randomized hyper-parameter search with 5-fold cross-validation to achieve  high estimation accuracy with improved generalization and robustness of targeted  variables in the black-box inverter model and black-box inverter compensation scheme.

  1. The description of the algorithms used cannot be followed, although this would be a crucial part of an otherwise 29-page article, which could be described as voluminous. I consider it a serious error to omit the various figures and numerical values by omitting the original source code. In my opinion, the work can be evaluated by bringing the ML algorithms to open platform, which can only be achieved by uploading this source code to GitHub. In this case, the reader can follow the algorithms

Answer: In this investigation the authors have used Python programming language with 1.0.2 version of scikit-learn library (available from December 2021). The scikit-learn library contains all the ML algorithms used in this research. So this was the base from which the authors started building up complex ensemble system with random hyper parameter grid search with 5-fold cross validation.

For basic “out of the box” approach in which the ML algorithms were used with default parameters these algorithms were simply called out from the library.

In case of 5-fold cross-validation with random hyper-parameter grid search the function for random hyper-parameter grid search had to be created for each ML algorithm. The 5-fold cross-validation was called from the scikit-learn library.

In case of stacking ensemble with random hyper-parameter grid search was developed for each base estimator (ML algorithm) and the 5-fold cross-validation was called from the scikit-learn library.

The authors agree with the reviewer's comment and have taken it into account. The descriptions of the used algorithms have been maximally reduced, keeping only basic information about ML algorithm since we did not developed any algorithm from scratch.  For more information about ML algorithms authors have provided additional reference to scikit-learn library (reference 40). However, the description of those and only those hyper-parameters that were used in the functions of random selection of hyper-parameters, as well as the table of their range, was left in the manuscript. By doing so we have reduced the tedious and detailed description of each ML algorithm and kept only the basic information.

In modified version of the manuscript (at the end of the manuscript) the authors have provided link to the GIT-HUB repository which can be used to check the developed Python scripts used in this research.

https://github.com/nandelic2022/InverterML/tree/main

  1. The thesis is not free of bugs and some serious errors (figures are illegible, it writes Hubber instead of Huber.

All figures are enlarged on average by 20 % and Hubber was corrected to Huber. We do hope that the figures are now legible.

Round 2

Reviewer 1 Report

The authors have addressed all my concerns.

Reviewer 2 Report

No further comments

Reviewer 3 Report

I accept the explanations and changes.